# Shipborne measurements of total OH reactivity around the Arabian Peninsula and its role in ozone chemistry

Eva Y. Pfannerstill[1], Nijing Wang[1], Achim Edtbauer[1], Efstratios Bourtsoukidis[1], John N. Crowley[1], Dirk Dienhart[1], Philipp G. Eger[1], Lisa Ernle[1], Horst Fischer[1], Bettina Hottmann[1], Jean-Daniel Paris[2], Christof Stönner[1], Ivan Tadic[1], David Walter[1], Jos Lelieveld[1,3], Jonathan Williams[1,3]

[1]Atmospheric Chemistry and Multiphase Chemistry Departments, Max Planck Institute for Chemistry, 55128 Mainz, Germany
[2]Laboratoire des Sciences du Climat et de l'Environnement, LSCE/IPSL, CEA-CNRS-UVSQ, Université Paris-Saclay, Gif-sur-Yvette, France
[3]Energy, Environment and Water Research Center, The Cyprus Institute, Nicosia, Cyprus

*Correspondence to*: Eva Y. Pfannerstill (eva.pfannerstill@mpic.de)

**Abstract.** The Arabian Peninsula is characterized by high and increasing levels of photochemical air pollution. Strong solar irradiation, high temperatures and large anthropogenic emissions of reactive trace gases result in intense photochemical activity, especially during the summer months. However, air chemistry measurements in the region are scarce. In order to assess regional pollution sources and oxidation rates, the first ship-based direct measurements of total OH reactivity were performed in summer 2017 from a vessel travelling around the peninsula during the AQABA (Air Quality and Climate Change in the Arabian Basin) campaign. Total OH reactivity is the total loss frequency of OH radicals due to all reactive compounds present in air and defines the local lifetime of OH, the most important oxidant in the troposphere. During the AQABA campaign, the total OH reactivity ranged from below the detection limit (5.4 s$^{-1}$) over the north-western Indian Ocean (Arabian Sea) to a maximum of $32.8 \pm 9.6$ s$^{-1}$ over the Arabian Gulf (also known as Persian Gulf) when air originated from large petroleum extraction/processing facilities in Iraq and Kuwait. In the polluted marine regions, OH reactivity was broadly comparable to highly populated urban centers in intensity and composition. The permanent influence of heavy maritime traffic over the seaways of the Red Sea, Gulf of Aden and Gulf of Oman resulted in median OH sinks of 7.9−8.5 s$^{-1}$. Due to the rapid oxidation of direct volatile organic compound (VOC) emissions, oxygenated volatile organic compounds (OVOCs) were observed to be the main contributor to OH reactivity around the Arabian Peninsula (9−35 % by region). Over the Arabian Gulf, alkanes and alkenes from the petroleum extraction and processing industry were an important OH sink with ~9 % of total OH reactivity each, whereas $NO_x$ and aromatic hydrocarbons (~10 % each) played a larger role in the Suez Canal, which is influenced more by ship traffic and urban emissions. We investigated the number and identity of chemical species necessary to explain the total OH sink. Taking into account ~100 individually measured chemical species, the observed total OH reactivity can typically be accounted for within the measurement uncertainty (50 %), with 10 dominant trace gases accounting for 20−39 % of regional total OH reactivity. The chemical regimes causing the intense ozone pollution around the Arabian Peninsula were investigated using total OH reactivity measurements. Ozone vs. OH reactivity relationships were found to be a useful tool for differentiating between ozone titration in fresh emissions

and photochemically aged air masses. Our results show that the ratio of $NO_x$- and VOC-attributed OH reactivity was favorable for ozone formation almost all around the Arabian Peninsula, which is due to $NO_x$ and VOCs from ship exhausts and, often, oil/gas production. Therewith, total OH reactivity measurements help to elucidate the chemical processes underlying the extreme tropospheric ozone concentrations observed in summer over the Arabian Basin.

## 1    Introduction

The Arabian Peninsula is a hotspot of global change. It is already subject to extremes of heat and drought, which are predicted to intensify (Zhang et al., 2005; Wasimi, 2010; Chenoweth et al., 2011; Tanarhte et al., 2012; Lelieveld et al., 2012; Terink et al., 2013; Donat et al., 2014; Cook et al., 2016; Lelieveld et al., 2016), and to photochemical air pollution (Lelieveld et al., 2009; Smoydzin et al., 2012; Abdelkader et al., 2015; Farahat, 2016). The Middle East accommodates densely populated urban centers with a total of 350 million people, and is a hub of fossil fuel extraction and processing with more than half of the world's known oil and gas reserves (Khatib, 2014). Moreover, the waters surrounding the Arabian Peninsula are key routes of global trade and bottlenecks for marine traffic, another source of air pollution (Endresen, 2003; Eyring, 2005; Boersma et al., 2015; Mertens et al., 2018) that is expected to increase in the future (Eyring et al., 2007).

Photochemistry in the region is thought to be particularly active due to a combination of factors: large anthropogenic emissions of reactive hydrocarbons and $NO_x$ (= $NO$ + $NO_2$) from the petroleum industry, shipping and energy-intensive urban areas; with high levels of solar irradiation and hence photolysis rates; and high temperature, which disfavors $NO_x$ reservoirs such as PAN and $N_2O_5$. The main regional photochemical oxidation process is initiated by ozone photolysis to $O(^1D)$, which reacts with water vapor to form hydroxyl radicals (OH). The OH radicals react with volatile organic compounds (VOCs) to yield peroxy radicals that can convert $NO$ to $NO_2$, leading to ozone formation (Haagen-Smit, 1952; Sillman, 1999; Ren et al., 2013). The abundance of precursors and intense photochemistry make conditions in the Arabian Basin very favorable for photochemical ozone production, leading to extremely high ozone mixing ratios up to 200 ppb (Lelieveld et al., 2009; Smoydzin et al., 2012; Krotkov et al., 2016). Ozone formation was reported to be enhanced even 300 km downwind of a hydrocarbon processing facility at the coast of the Arabian Gulf (Moradzadeh et al., 2019). In addition, VOC oxidation by OH and ozone can generate secondary organic aerosol (Grosjean and Seinfeld, 1989; Kroll and Seinfeld, 2008) with implications for human health (Lelieveld et al., 2015) and regional radiative budgets (Ezhova et al., 2018).

Despite being an air pollution hotspot, very few observational atmospheric data exist from the Arabian Basin, and studies of VOCs in the region are scarce and mostly urban (Doskey et al., 1999; Matysik et al., 2010; Salameh et al., 2014; Simpson et al., 2014; Khalil et al., 2016; Salameh et al., 2016; Barletta et al., 2017). In order to chemically map the region, a comprehensive suite of atmospheric measurements was performed from a ship sailing around the Arabian Peninsula in summer 2017 during the AQABA (Air Quality and Climate Change in the Arabian Basin) cruise.

The number of known VOCs in air has been estimated to be between $10^4$ and $10^5$ species (Goldstein and Galbally, 2007), and it is assumed that many more remain unknown. Accounting for all of these chemical compounds individually with quantitative measurements is currently unfeasible. Instead, the combined load of OH reactive species (mainly VOCs and CO) in ambient air, termed the total OH reactivity, can be measured directly. Total OH reactivity denotes the total sink, or

loss frequency, of the OH radical. When compared to the OH reactivity calculated from all individually measured compounds (termed the speciated OH reactivity), total OH reactivity can be used to infer the occurrence of VOCs that were not measured (termed the unattributed or "missing" OH reactivity). Moreover, total OH reactivity is necessary for a quantitative understanding of ozone production and can help estimating its sensitivity to VOCs and $NO_x$ (Kirchner et al., 2001; Sinha et al., 2010; Pusede et al., 2014; Yang et al., 2017). The OH sink also affects the atmospheric residence time of

methane and therewith its global budget, which still poses a challenge in models (Zhao et al., 2019).

With VOC and $NO_x$ emissions from ship exhausts (Boersma et al., 2015; Endresen, 2003; Huang et al., 2018) and VOCs from oil and gas operations (Buzcu and Fraser, 2006; Gilman et al., 2009; Gilman et al., 2013; Schade and Roest, 2016; Milazzo et al., 2017), the air around the Arabian Peninsula is expected to contain significant levels of pollution. In Saudi Arabian cities, OH reactivity has been reported to range from 14 to 42 $s^{-1}$ (from the sum of individual measurements),

dominated by alkenes (Barletta et al., 2017). Downwind of oil and gas production in Colorado, OH reactivity of VOCs was dominated by alkanes (60 %), followed by OVOCs (27 %) and alkenes (8 %; Gilman et al., 2013). The largest OH sink among the VOCs in the exhaust plume of a ship operating at full speed are alkenes (42 %), followed by aromatics (40 %), OVOCs (12 %) and alkanes (5 %; estimated using emission factors from Huang et al., 2018).

Unattributed OH reactivity in locations dominated by anthropogenic VOCs has been measured to be as low as zero in Lille,

France (Hansen et al., 2015) or New York, U.S. (Ren et al., 2006), to 10−54 % in Paris, France (Dolgorouky et al., 2012) and 10−80 % in Tokyo, Japan (Yoshino et al., 2006). However, comparing unattributed OH reactivity between different studies is difficult, because it depends on the air mass and coverage of inorganic trace gases and VOCs measured in each campaign. In this work, we apply an approach that aims to use comprehensive trace gas information gathered by various instruments, even compounds with unidentified chemical structure measured by PTR−ToF−MS, for calculation of speciated OH

reactivity. Therewith, we identify the trace gases representing the major OH sinks and try to obtain closure on apportioning OH reactivity.

This study provides an overview of total OH reactivity around the Arabian Peninsula and investigates its sources in the different environments along the route of the campaign. Total OH reactivity is applied for assessing the regional oxidation chemistry and the underlying reasons for the extreme ozone concentrations observed in summer over the Arabian Basin. By

combining OH reactivity and ozone measurements, we aim to investigate ozone production processes and to identify differentiated chemical regimes. We demonstrate the utility of in-situ OH reactivity measurements to characterize the role of anthropogenic emissions in modifying the chemical composition of the air.

## 2 Materials and methods

### 2.1 AQABA Campaign

The Air Quality and Climate Change in the Arabian Basin (AQABA) measurements took place on-board the vessel *Kommandor Iona* from June 25 till September 3, 2017. The first leg began in La-Seyne-sur-mer (near Toulon, France) continued via the Mediterranean, the Suez Canal, the Red Sea, the Indian Ocean, the Gulf of Oman and the Arabian Gulf to Kuwait. On the second leg, the vessel returned via the same route (Fig. 1a/b). The ship was equipped with a weather station and four laboratory containers with instrumentation for in-situ and offline monitoring of a large suite of (trace) gases, particles and radicals.

### 2.2 Sampling

Air sampling was performed from a high-flow ($10\ \mathrm{m^3\ min^{-1}}$) cylindrical stainless steel inlet (sampling height: 5.5 m above deck, diameter: 0.2 m), situated between the containers on the front deck of the ship. This position ensured that inlets were ahead of any contamination sources from the vessel itself when pointed into the wind. Air was drawn from the center of the high-flow inlet and sub-sampled to the VOC container at a rate of approximately 5 slpm (standard L $\mathrm{min^{-1}}$, first leg) and 3 slpm (second leg) into an air conditioned laboratory container using a FEP (fluorinated ethylene propylene) inlet tube (1/2" = 1.27 cm o.d., length ca. 10 m), heated to 50−60 °C. A PTFE (polytetrafluoroethylene) filter, changed weekly (and additionally after intense particle contamination events such as dust storms), prevented contamination of the sub-sampling line by particles or sea spray. This inlet system was used for monitoring both VOCs and total OH reactivity simultaneously. We assume that the length of the inlet prevented some of the larger, extremely low volatile VOCs from reaching the instruments (see Sect. 3.3). The combined inlet residence time in FEP tubing and reactor for the OH reactivity instrument was determined as $\approx 15$ s during the first leg and $\approx 25$ s during the second leg by spiking with acetone.

### 2.3 OH reactivity and trace gas measurements

Total OH reactivity was measured using the Comparative Reactivity Method (CRM, Sinha et al., 2008), which proved to be robust enough to run nearly continuously for 10 weeks on a ship, often in rough conditions due to high waves. The method has been applied in many regions of the world (Yang et al., 2016) and has recently been compared to other methods (Fuchs et al., 2017). The CRM is based on a competitive reaction between reactive compounds from ambient air and a reagent, pyrrole (Westfalen AG, Münster, Germany), inside a glass reactor. OH radicals are produced in the reactor by flushing humidified nitrogen (6.0 grade, Westfalen AG, Münster, Germany) over a Hg/Ar UV lamp (LOT Quantum Design, Darmstadt, Germany). CRM uses three different modes: C1 (pyrrole + OH scavenger + UV light, at ≈ ambient humidity), C2 (OH + pyrrole, ambient humidity), and C3 (ambient air + pyrrole + OH). For more details of the method see Sinha et al. (2008) and Michoud et al. (2015). The system was operated at a pyrrole/OH ratio of $1.9 \pm 0.1$ (average ± standard deviation). Because this ratio deviates from pseudo-first order conditions, as is typical for CRM, a correction needed to be applied (see

Sect. 2.4). Sampling air was continuously drawn into the system and the instrument was switched between C3 (measured for 22-24 minutes) and C2 (measured for 6-8 min) modes. The longer C2 measurements were used when ambient humidity was unstable. C1 was determined every 5−7 days (average ± standard deviation over the whole campaign: 60.71 ± 1.18 ppb of pyrrole). Changes in pyrrole mixing ratio were monitored by a Proton Transfer Reaction-Quadrupole Mass Spectrometer (PTR−QMS, Ionicon Analytik, Innsbruck, Austria (Lindinger and Jordan, 1998) at $m/z = 68$ (dwell time: 5 s). The PTR−QMS was calibrated every week. It was operated at 60 °C drift temperature, 2.2 mbar drift pressure and 600 V drift voltage ($E/N$ = 137 Td). In parallel, a PTR−Time of Flight (ToF)−MS was deployed to monitor VOCs and OVOCs ($m/z$ up to 280), and a gas chromatograph with flame ionization detector (GC−FID) to monitor non-methane hydrocarbons (NMHCs; 20 compounds). Average detection limits for the different NMHCs ranged from 1−25 ppt. The NMHC and methane measurements are described in detail elsewhere (Bourtsoukidis et al., 2019). The PTR-ToF-MS was deployed at 60 °C drift temperature, 2.2 mbar drift pressure and 600 V drift voltage ($E/N$ = 137 Td). 1,3,5-trichlorobenzene was fed continuously into the sample stream for mass scale calibration. The time resolution of the measurement was 1 min and background measurements were performed every 3 h for 10 min. The PTR-ToF-MS was calibrated with a multicomponent pressurized gas VOC standard (Apel-Riemer Environmental Inc., Colorado, USA). Mass resolution (full width at half maximum) at 96 amu ranged between ~3500 and ~4500. Average detection limits for the compounds measured by PTR-ToF-MS ranged from 1−107 ppt. A full list of the trace gases measured by GC-FID and PTR-ToF-MS can be found in Table S1.

Formaldehyde (HCHO) and inorganic trace gases (NO, $NO_2$, $O_3$, $SO_2$) were measured sampling from the same chimney (the instruments used are specified in Table S1). Based on $SO_2$, $NO_x$ and ethene measurements, a filter was applied to exclude sampling periods that were influenced by the vessel's own exhaust.

## 2.4 Total OH reactivity data analysis

CRM data analysis was conducted following the general procedures described in Keßel (2016) and Pfannerstill et al. (2018). Humidity in C2 (background) measurements can differ from that of C3 (ambient) measurements because C2 was measured only twice per hour. As the amount of OH radicals generated in the CRM reactor depends on humidity, C2 data were corrected for these C2/C3 humidity differences by applying an empirical relationship of the pyrrole signal vs. the PTR−MS water cluster signal as described in Michoud et al. (2015). The C2 correction applied to match C3 humidity decreased pyrrole levels by on average 0.05 ± 0.04 ppb, corresponding to an OH reactivity of $\approx$ 0.3 s$^{-1}$. NO, $NO_2$ and ozone cause interferences in CRM due to OH recycling induced by $HO_2$, which is formed in the reactor simultaneously with OH (Sinha et al., 2008; Michoud et al., 2015; Fuchs et al., 2017). Ozone, NO and $NO_2$ interferences were quantified in the laboratory (details of the experiments can be found in Sect. S1). $NO_2$ loss and conversion to NO in the mass flow controller during tests was monitored by an NO/$NO_2$ analyzer (Two-channel-chemiluminescence detector). The combined NO, $NO_2$ and ozone corrections amounted to a median of ~24 % of total OH reactivity or an average of 0.36 ± 0.19 ppb of pyrrole over the whole campaign, which equals $\approx$ 1.9 s$^{-1}$, with a maximum absolute correction of 1.39 ppb (7.4 s$^{-1}$ / 27 % of total OH reactivity)

under the influence of closely located ship emissions (high $NO_x$). A frequency distribution of the corrected fraction of total OH reactivity is displayed in Fig. S1.

The calculation of total OH reactivity with the CRM equation (see Sinha et al., 2008) assumes pseudo-1st-order conditions inside the reactor, which can, however, not be met while preserving a reasonable sensitivity (C3−C2 difference). The correction for deviation from pseudo-1st-order conditions inside the reactor was derived from empirical test gas measurements adapted from the approach in Michoud et al. (2015). The deviation from pseudo-1st-order depends on the pyrrole/OH ratio and on the reaction rate constant of the measured trace gas towards OH. Trace gases with fast reaction rates (comparable to that of pyrrole) require the largest correction because their concentration inside the reactor decreases at a similar rate as that of pyrrole, leading to an underestimation of their reactivity. Such gases, namely alkenes, were sometimes present during AQABA (e.g. a $C_5H_8$-alkene which was not isoprene). This is why we did not use an average correction factor for all gas mixes as in Michoud et al. (2015), but applied a correction where the composition of the air mass is taken into account. Test gases used were toluene (for aromatics correction), isoprene and propene (for alkenes correction), propane and cyclohexane (correction for other species). These test gases are representative of the relevant trace gases and ranges of reaction rate coefficients observed during AQABA. In test experiments, these trace gases were added to the CRM system in known amounts, i.e. known total OH reactivities, at pyrrole/OH ratios representative for the campaign. The correction factors resulting from comparing the observed reactivity with the expected reactivity were weighted according to the measured composition of the ambient air at the moment of observation:

$$F = a\ X_{aromatic} + b\ X_{alkenes} + c\ (1 - X_{arom} - X_{alkenes}) \qquad \text{Eq. (1),}$$

where $F$ is the total correction factor and the weighted compound-class-specific correction factors are a = 1.22, b = 2.89 and c = 1.44. $X$ is the speciated OH reactivity fraction (in $s^{-1}$ s) of the respective class of compounds (aromatics, alkenes). This correction increased OH reactivity by a factor of 1.4 to 2.5 (average ± standard deviation: $F = 1.5 \pm 0.1$, median = 1.5).

The dilution of ambient air with humidified nitrogen was accounted for with a dilution factor of 1.34.

The 5 min detection limit (LOD) was 5.4 $s^{-1}$, derived from the $2\sigma$ standard deviation of clean air measurements in the Arabian Sea. Total uncertainty (1 $\sigma$) of the measurements was ~ 50 % (median and average), with a precision of 10−76 % over 5 min depending on the quantity of reactivity.

## 2.5 Calculated OH reactivity from individually measured compounds (speciated OH reactivity)

The speciated OH reactivity is the sum of the OH reactivities attributed to individual (measured) trace gases:

$$R = \Sigma\ k_{M+OH}\ [M] \qquad \text{Eq. (2)}$$

Contributions of trace gases (M, where $[M]$ is their respective concentration in molecules $cm^{-3}$) to OH reactivity ($R$ in $s^{-1}$) are calculated using the reaction rate constants ($k_{M+OH}$ in $cm^3$ molecule$^{-1}$ $s^{-1}$) of the respective trace gases (M) with the OH radical. The difference between measured total OH reactivity and the sum of individual trace gas contributions to OH reactivity is termed "unattributed" or "missing" OH reactivity. The trace gases taken into account here for calculating speciated OH reactivity and their reaction rate constants used are listed in Table S1. Out of a total of 120 chemical species

that were considered for the calculation of speciated reactivity, 42 chemical species (specified in Table S1), including many of the known important contributors to OH reactivity, were calibrated with gas standards and therefore have low uncertainties in their concentrations as well as in their reaction rate coefficients (5−15 %). A further 78 exact masses (specified in Table S1) monitored by PTR−ToF−MS were attributed to molecular formulae, and their concentrations derived using a theoretical approach (Lindinger and Jordan, 1998), which has an uncertainty of ca. 50 %. In cases where several trace gases could be responsible for the measured mass, an average of the known reaction rate coefficients with OH was used for calculating the speciated OH reactivity. In the few cases where rate coefficients were unknown in literature, the rate coefficient of a trace gas with comparable functional groups was applied (see Table S1). The uncertainty of the reaction rate coefficients $k_{M+OH}$ is estimated to be 100 % due to the occasional large differences in rate coefficients between possible structures. The uncertainty of the resulting speciated (i.e. calculated) OH reactivity depends on the fraction of gas-standard calibrated compounds in the ambient air at any given point of time and varies between 10 % and 92 %, with an average and median of 45 % over the whole campaign. In accordance with the time resolution of the GC-FID, which monitored compounds with a significant share of OH reactivity, speciated OH reactivity was calculated in 50 min time resolution.

## 2.6 Air mass back trajectories

To investigate the origin of air masses encountered, back trajectories were derived using the Hybrid Single-Particle Lagrangian Integrated Trajectory model (HYSPLIT, version 4, 2014). This model is a hybrid between a Lagrangian and an Eulerian model for tracing small imaginary air parcels forward or back in time (Draxler and Hess, 1998). Back trajectories were calculated based on a start height of 200 m above sea level, going 216 h back in time on an hourly grid beginning at the ship position.

## 3   Results and discussion

### 3.1 Overview of total OH reactivity around the Arabian Peninsula

The air masses encountered during the AQABA campaign were delineated according to the geographical region (Fig. 1a/b). A notable difference between leg 1 and leg 2 was the general wind direction over the Arabian Gulf, which changed from north-west (from Iraq/Kuwait) to northeast (from Iran). The range of observed total OH reactivities over the whole campaign was from below the detection limit over the Arabian Sea (north-west Indian Ocean) up to $303.6 \pm 83.9$ s$^{-1}$ during fueling/bunkering of the ship at the port of Fujairah (United Arab Emirates), due to fresh fuel emissions. The largest fraction of attributed OH reactivity was provided by OVOCs in almost all AQABA regions (9−35 %), which is comparable to previous studies in the oil impacted region of Houston, where between 11 % and 24 % of the OH sink was due to OVOCs (Mao et al., 2010). There were notable regional differences in both the total OH reactivity and the contributing trace gases, which will be discussed hereafter. An overview is displayed in Table 1, which divides the reactivity into different chemical families. A list of chemical species attributed to each chemical family is available in Table S1.

The measured sulfur-containing VOCs (SVOCs) and halogenated VOCs (HalVOCs) played virtually no role for OH reactivity in any of the AQABA regions (Table 1). The OH reactivity of the monitored NVOCs (nitrogen-containing VOCs) was also very minor with not more than 1 % of total OH reactivity over the whole campaign.

### 3.1.1 Strongly polluted regions: Arabian Gulf, Gulf of Suez and Suez Canal

Compounds with short lifetimes are generally more relevant for total OH reactivity than long-lived species (see also Sect. 3.3). Since emissions from oil and gas production and cities contain high mixing ratios of reactive alkenes, the Arabian Gulf, the Gulf of Suez and the Suez Canal, where such sources abound, showed the highest average OH reactivities (Fig. 1c/d). In these regions, obvious nearby emission sources were cities, industrial complexes, and especially oil/gas production/processing (Fig. 2). The large alkane mixing ratios associated with oil and gas production (Fig. 1e; Bourtsoukidis et al., 2019) were not reflected in an equally large share of total OH reactivity due to the slow reaction rate coefficient of alkanes with the OH radical (Fig. 1d). The average total OH reactivity (± standard deviation) was $12.9 \pm 6.2$ s$^{-1}$ (median: 11.2 s$^{-1}$) over the Arabian Gulf, $13.2 \pm 6.9$ s$^{-1}$ (median: 10.8 s$^{-1}$) over the Suez Canal and $11.6 \pm 4.2$ s$^{-1}$ (median: 10.4 s$^{-1}$) over the Gulf of Suez. There were short periods with higher total OH reactivities (± total uncertainties) up to $32.8 \pm 9.6$ s$^{-1}$ over the Arabian Gulf and $26.6 \pm 8.2$ s$^{-1}$ over the Suez Canal (see Sect. 3.2). These ranges are comparable to OH reactivities reported from some urban areas, e.g. Beijing, China (Williams et al., 2016), Houston, Texas (Mao et al., 2010), New York City (Ren et al., 2006), and to OH reactivities calculated from ship-based VOC observations in the fossil fuel production impacted Houston/Galveston Bay (Gilman et al., 2009). The AQABA observations are, however, more than three times as high as VOC-attributed OH reactivity calculated from observations downwind of gas and oil production in south Texas (Schade and Roest, 2016) and twice as high as at the Boulder Atmospheric Observatory when impacted by gas wells (Swarthout et al., 2013).

The largest percentage of attributed OH reactivity over the Arabian Gulf was provided by OVOCs (35 % of total, in contrast to 14 % observed over the Houston/Galveston Bay (Gilman et al., 2009). This was surprising, as the Gulf region was expected to be similarly dominated by direct hydrocarbon emissions from the fossil fuel industry as the Houston/Galveston Bay. Although the alkanes and OVOCs had broadly equivalent mixing ratios over the Arabian Gulf (Fig. 1e), the former have generally lower OH reactivity. Conditions in the Gulf region during summer favor rapid photooxidation of hydrocarbon emissions to OVOCs, due to high temperatures (on average 34.5 °C), high humidity (up to 92 % RH) and strong irradiation. It should also be noted that as the ship passed through the Arabian Gulf, it was only exposed to primary emissions for relatively short periods of time, when located directly downwind of the sources. The more oxidized regional background air was sampled for longer periods. Furthermore, a bias towards OVOCs in the assessment of speciated OH reactivity is also possible in this study, because they were well covered by PTR−ToF−MS measurements, while longer-chain or branched alkenes and alkanes, which are components of oil and other fuels (D'Auria et al., 2009; Gueneron et al., 2015) were in the AQAQBA measurements limited to $C_2$-$C_8$ compounds. Nonetheless, considering the whole campaign, the Arabian Gulf was

the region with the highest OH reactivity from alkanes (on average ~1.2 s$^{-1}$ or ~9 % of total OH reactivity). An important influence of oil and gas production on total OH reactivity in the Arabian Gulf is therefore evident in our data.

Petroleum extraction facilities and refineries are located at both shores of the Gulf of Suez and Canal, (Fig. 2), and wind from both east and west lead to a relatively large contribution of alkanes (5−6 %; Fig. 1d/e).

By virtue of their double bond, alkenes are more reactive towards OH than alkanes. Indeed, in Saudi Arabian cities, OH reactivity of alkenes has been reported to be twice as high as the reactivity of alkanes, despite higher abundance of the latter (Barletta et al., 2017). Similarly, over the Arabian Gulf, the AQABA campaign results show an equivalent average share of OH reactivity from alkenes and alkanes (both ~1.2 s$^{-1}$, which equals ~9 %), despite a ~11 times larger average mixing ratio of alkanes. Certainly a fraction of highly reactive alkenes emitted on land will have already been oxidized before the

emission impacted air mass reached the point of observation on the ship. Alkene-attributed OH reactivity was slightly higher in the Suez Canal (1.4 s$^{-1}$ vs. 1.2 s$^{-1}$), where the distance to emission sources is generally shorter due to the narrowness of the canal. Additionally, and probably more importantly, alkenes are only minor components in fossil fuel emissions generally. They are emitted from vehicle exhausts (a possible source near the Suez Canal) due to incomplete combustion (Koppmann, 2007). Alkanes, on the other hand, can be so abundant in emissions from oil and gas production that they dominate OH

reactivity, as was observed in Colorado, where 60 % of OH reactivity calculated from VOC measurements (not total OH reactivity) was due to alkanes and only 8 % from alkenes (Gilman et al., 2013). In another study at the same site, alkanes contributed 26 %, alkenes 3 %, and aromatics 2 % of total OH reactivity (Swarthout et al., 2013). In contrast to that, alkenes (41 %) dominated over alkanes (23 %) in the OH reactivity calculated from VOC observations over the Houston/Galveston Bay (Gilman et al., 2009).

Aromatics are important components of fossil fuels besides alkanes and alkenes (Koppmann, 2007). In Egyptian cities, ambient VOC mixing ratios from traffic have been reported to be dominated by aromatics (Matysik et al., 2010), whereas in Saudi Arabian cities, alkanes were more abundant (Barletta et al., 2017). Consistent with these findings, we observed higher average speciated OH reactivity from aromatics (1.3 s$^{-1}$) in the Suez Canal (influenced by air from Cairo, roads next to the Canal, a refinery, and a power plant) compared to the Arabian Gulf (0.9 s$^{-1}$, influenced by oil/gas industry emissions and

partly by urban air from Saudi Arabia, Iraq and Kuwait).

A strong influence of maritime traffic in the narrow and highly frequented Suez Canal and Gulf is evident from the large NO$_x$ contribution to total OH reactivity (~10 and ~12 %, respectively). Here, other vessels and their exhaust plumes passed by closely during waiting periods (before entering the Suez Canal in the northern Gulf of Suez and inside the Canal).

The Arabian Gulf and Suez Canal will be discussed in more detail in case studies in Sect. 3.2.

**3.1.2 Seaways influenced by maritime traffic: Red Sea, Gulf of Aden and Gulf of Oman**

The Red Sea, the Gulf of Aden, and the Gulf of Oman displayed similar total OH reactivities with medians of 7.9−8.5 s$^{-1}$ during the AQABA campaign (northern Red Sea: median 8.5 s$^{-1}$ / mean ± standard deviation 9.1 ± 3.3 s$^{-1}$, southern Red Sea:

7.9 $s^{-1}$ / 8.7 $\pm$ 4.2 $s^{-1}$, Gulf of Aden: 8.0 $s^{-1}$ / 8.2 $\pm$ 3.1 $s^{-1}$, Gulf of Oman: 8.4 $s^{-1}$ / 8.4 $\pm$ 3.1 $s^{-1}$, also see Table 1 and Fig. 1c/d). These values are comparable to urban observations in the lower range, such as in in Helsinki, Finland (Praplan et al., 2017), or Mainz, Germany (Sinha et al., 2008).

The OH reactivity over the Red Sea, Gulf of Aden and Gulf of Oman can largely be attributed to the influence of maritime transport in these busy, relatively constricted seaways. Refineries at the eastern shore of the northern Red Sea (Fig. 2) are not thought to have influenced our measurements due to prevailing wind from north-west and west.

Among the three regions, the Gulf of Aden was the one with the lowest average total OH reactivity. Likely, this is due to its location in the transition zone to the open north-western Indian Ocean (see Fig. 1c). Here, air masses were less influenced by pollution than over the Red Sea or Gulf of Oman, as the AQABA cruise took place during northern hemisphere summer, when the ITCZ is over India and comparably clean air is drawn north over the north-western Indian Ocean (Ajith Joseph et al., 2018). In accordance with this finding, Bourtsoukidis et al. (2019) showed by lifetime-variability regression of alkanes that the Gulf of Aden was the area with the most remote character during AQABA.

In ship exhausts, the highest emission factors (apart from $CO_2$) have been found for $NO_x$ (Endresen, 2003; Eyring, 2005; Huang et al., 2018). Large ship engines emit an average of 76 kg of $NO_x$ per ton of fuel combusted (Eyring, 2005). This is reflected in the OH reactivity share of $NO_x$, which was in these relatively narrow seaway regions in the range of 4−9 % (Table 1). OVOCs accounted for 12−21 % of total OH reactivity there, alkenes 4−9 %, inorganics (dominated by $SO_2$ and CO) 4−6 %, aromatics 3−8 % and alkanes 4−5 %. All those classes of compounds can stem from fuel combustion in ships (Endresen, 2003; Eyring, 2005; Huang et al., 2018; Xiao et al., 2018), which emit an average of 43 kg of $SO_x$, 4.7 kg of CO and 7 kg of hydrocarbons per ton of fuel combusted (Eyring, 2005).

The $C_3$ carbonyls ($C_3H_6O$) vs. propane ($C_3H_8$), $C_5$ oxidation products ($C_5H_{10}O$) vs. pentanes ($C_5H_{12}$) and toluene vs. benzene ratios (Table 1) were used here as indicators of the oxidation state (and, thereby, the extent of photochemical ageing) of the air. We note that this approach is based on two assumptions: 1) that production of the carbonyls by oxidation dominates over direct emission, although some direct combustion emissions cannot be excluded, and 2) that the only carbonyl precursor is the corresponding $C_3$ or $C_5$ compound without significant fragmentation of larger VOCs. The toluene vs. benzene relationship is based on the assumption of simultaneous emission of both compounds. Notably, a distinct difference between northern (less oxidized, $[C_3H_6O]/[C_3H_8] \approx 1.9$ and $[C_5H_{10}O]/[C_5H_{12}] \approx 1.3$) and southern Red Sea (oxidized, $[C_3H_6O]/[C_3H_8] \approx 9.4$ and $[C_5H_{10}O]/[C_5H_{12}] \approx 1.9$) was observed. Although it did not result in large differences in total OH reactivity values (median of 7.9 $s^{-1}$ in the photochemically aged air masses in the south Red Sea and 8.5 $s^{-1}$ in the north Red Sea), it reflects the origin of air masses: urban centers in northeast Africa for the north Red Sea, and the less populated and more distant Sudan and Chad for the south Red Sea.

### 3.1.3 Cleaner regions: Mediterranean and Arabian Sea

Over the more open seas, notably the Mediterranean and Arabian Sea (north-western Indian Ocean), the air was aged, cleaner, and less reactive to OH. Here, total OH reactivities were mostly close to the detection limit of 5.4 $s^{-1}$. The median

over the Mediterranean was 6.8 s$^{-1}$ (average $\pm$ standard deviation: 7.2 $\pm$ 2.9 s$^{-1}$), which compares to the lower limits of observations from suburban areas such as Whiteface Mountain (Ren et al., 2006), Norfolk, UK (Ingham et al., 2009), Central Pennsylvania (Ren et al., 2005) or Arenosillo at the Spanish coast (Sinha et al., 2012), and more remote locations such as the Rocky Mountains (Nakashima et al., 2014). Total OH reactivity measured over the Arabian Sea during AQABA was with a

median of 4.9 s$^{-1}$ (average $\pm$ standard deviation: 5.6 $\pm$ 2.0 s$^{-1}$) below the detection limit of 5.4 s$^{-1}$. The marine OH reactivity calculated from trace gas measurements over the central Gulf of Mexico was 1.01 s$^{-1}$ (Gilman et al., 2009), a value which was slightly exceeded in our observations over the open Arabian Sea with a speciated OH reactivity of 1.5 $\pm$ 0.2 s$^{-1}$.

Larger OH reactivity occurred over the Mediterranean Sea during short episodes of polluted air from the mainland in the Strait of Messina (between Sicily and Italy), visible in Fig. 1b. In the Strait of Messina and surroundings, total OH reactivity

amounted to between 6.7 $\pm$ 3.6 s$^{-1}$ and 26.7 $\pm$ 8.2 s$^{-1}$ for 5 h, whereas before and after, it was close to the detection limit. OVOCs were with ~11$-$12 % of total OH reactivity the largest group of identified OH sinks in both the Mediterranean and Arabian Sea. Over the Mediterranean, the influence of shipping emissions was larger than over the Arabian Sea, evident from NO$_x$-attributed OH reactivity, which was on average 0.26 s$^{-1}$ or ~4 % of the total in the Mediterranean in contrast to 0.05 s$^{-1}$ or ~1 % of the total in the Arabian Sea. Johansson et al. (2017) found that ship traffic over the Arabian Sea causes

3 % of the global NO$_x$ emissions from maritime transport, which explains why even here the conditions were not completely pristine.

### 3.2 Case studies

### 3.2.1 Arabian Gulf

Figure 1 shows the average OH reactivity composition for the Arabian Gulf. However, the air measured over the Arabian

Gulf originated partly from Kuwait/Iraq, partly from Iran and partly from the south. The composition of total OH reactivity between the different regions of origin displayed distinct differences, and can therefore be divided according to origin of the measured air masses, shown in Fig. 3.

Notably, the air from Iraq and/or Kuwait showed obvious influence from oil extraction industries with large fractions of alkanes, alkenes and aromatics (Fig. 3). Here, emission sources were closest to the point of observation, which was reflected

in highest total OH reactivity (median: 18.8 s$^{-1}$, average $\pm$ standard deviation: 18.7 $\pm$ 7.9 s$^{-1}$) and low C$_3$ carbonyls/propane and high toluene/benzene ratio medians of 0.3 and 0.8, respectively. In contrast, OH reactivity in the air originating from Iran was lower (median: 9.8 s$^{-1}$, average $\pm$ standard deviation: 10.8 $\pm$ 3.4 s$^{-1}$), with a slightly higher C$_3$ carbonyls/propane and a lower toluene/benzene ratio (both 0.4). The OH reactivity here was similar to that found in air from the south (Saudi Arabia) with a median of 9.8 s$^{-1}$ and a mean of 9.7 $\pm$ 3.3 s$^{-1}$, however, the C$_3$ carbonyls/propane ratio of 2.0 points to more

oxidized air masses. The high OH reactivity in air originating from Kuwait and Iraq can be attributed to emissions from petrochemical extraction and processing industries that would explain the large contribution of alkanes, alkenes and aromatics. Apart from two notable exceptions, the distribution of OH reactivity did not differ significantly between air from

Iran and the south. In air from Iran, the alkene fraction was larger than in air from the south, probably due to closer and different emission sources (gas extraction at the Iranian coast), which would also explain the lower $C_3$ carbonyls/propane ratio. The $NO_x$ fraction was larger in air from south, most likely for reasons of ship traffic influence.

Figure 4 (a) depicts a time series of measurements along the ship track from entering the Arabian Gulf to Kuwait and back.

The upper part of the graph shows indicators of the photochemical age of the air mass. The $[C_3H_6O]/[C_3H_8]$ ratio decreases and the toluene/benzene ratio increases when getting closer to Kuwait, meaning that the research vessel was approaching the emission sources (refineries, oil platforms). Both these indicators show opposite patterns, as expected. The $[C_5H_{10}O]/[C_5H_{12}]$ ratio follows the $[C_3H_6O]/[C_3H_8]$ ratio except for an increase towards Kuwait, which shows that the fresh emissions observed here were richer in propane than in pentanes, and indicates a mix of fresh emissions with aged air masses. This observation

will be discussed in more detail in Sect. 3.4.2. Changes in relative humidity indicate the varying influence of dry desert air and more humid air masses.

The following five case studies (orange numbered labels in Fig. 4 with corresponding air mass back trajectories) will go into more detail regarding the origin of OH reactivity (reported with ± 1 σ total uncertainty) in the Arabian Gulf:

1) At the entrance to the Arabian Gulf after passing the Strait of Hormuz, total OH reactivity was comparably low

$(8.0 \pm 4.0 \text{ s}^{-1})$. Sampled air had traversed the open Indian Ocean, but passed over the Musandam Peninsula, where urban areas probably contributed some OH reactivity. OVOCs dominated $(36 \pm 20 \text{ \%}$ of OH reactivity), with an insignificant unattributed fraction of $38 \pm 76 \text{ \%}$.

2) At this location in the northern part of the Arabian Gulf, ca. 100 km from the coast of Saudi Arabia, total OH reactivity amounted to $22.1 \pm 7.2 \text{ s}^{-1}$, mainly due to OVOCs $(57 \pm 29 \text{ \%})$, with large fractions of alkanes $(10 \pm 1 \text{ \%})$, alkenes $(16 \pm 8 \text{ \%})$

and aromatics $(10 \pm 5 \text{ \%})$. The back trajectories originate from eastern Iraq, with the urban areas of Amara and Basra. The large fraction of OVOCs is an indicator for chemically aged air. The aromatics, alkenes and alkanes probably were emitted from offshore oil/gas extraction platforms (Fig. 4b) and mixed with the oxidized air contaminants coming from larger distance. Within the uncertainty of the measurement, there is no unattributed OH reactivity at this location; with values of $19.4 \pm 9.9 \text{ s}^{-1}$ for the speciated OH reactivity and $22.1 \pm 7.2 \text{ s}^{-1}$ for the measured total OH reactivitiy.

3) Approximately 30 km from the coast of Kuwait, the back trajectories indicate that the sampled air masses travelled over the Rumaila oil field in Basra (Iraq). Consequently, the total OH reactivity of $26.4 \pm 8.1 \text{ s}^{-1}$ was mainly due to alkanes from oil extraction $(24 \pm 2 \text{ \%})$. In contrast to the previous case study, significant unattributed OH reactivity was observed here $(40 \pm 37 \text{ \%})$, which might be due to components of oil such as higher than $C_8$ and/or branched hydrocarbons that were not measured on board the *Kommandor Iona*.

4) Just after departure from the port of Kuwait, the highest observed OH reactivity $(32.8 \pm 9.6 \text{ s}^{-1})$ of the campaign was measured (except for refueling). HYSPLIT back trajectories show that it likely originated from large pollution sources at the coast: refineries (Saudi Aramco refinery and industrial area with storage tanks in Khafji, Saudi Arabia) and oil fields (Wafra oil field in Kuwait; oil platforms at the sea). Notably, alkenes $(23 \pm 14 \text{ \%})$ were a much larger OH sink here than alkanes $(5 \pm 0.5 \text{ \%})$, and the unattributed OH reactivity was zero within the uncertainty.

5) The largest OH reactivity measured during the whole campaign was $303.6 \pm 83.9$ s$^{-1}$ during refueling of the vessel in front of the port of Fujairah (United Arab Emirates). A back trajectory would not be relevant here, as these are direct diesel fuel evaporation emissions, mainly composed of alkenes ($27 \pm 18$ % of speciated OH reactivity) and aromatics ($21 \pm 10$ %), mixed with NO$_x$ ($39 \pm 4$ %) from the bunker ship's own fuel combustion. Wu et al. (2015) measured total OH reactivity from gasoline evaporation with the CRM method. In their study, alkenes contributed 40 % of measured OH reactivity, and they found an unattributed OH reactivity fraction of 43 %. In our study, a larger percentage ($75 \pm 42$ %) of total OH reactivity remains unexplained in the fuel evaporation measurement, which is most likely due to the fact that less of the higher and branched hydrocarbons were measured.

### 3.2.2 Suez Canal

The Suez Canal transit on August 24, 2017, was among the episodes of highest total OH reactivities during the whole campaign and will therefore be discussed in more detail here. In August 2017, 49 ships transited the Canal per day (Suez Canal Authority, 2018). It is one of the world's most heavily used shipping lanes and there are no regulations concerning the fuel used or emission limitations. The Suez Canal is 193 km long and at the water surface 313 m wide (Suez Canal Authority, 2017). Given its narrowness, other vessels and their exhausts, as well as urban areas or streets were relatively close to the *Kommandor Iona* while it was traveling along the Canal. Consequently, variations in OH reactivity can mainly be attributed to local emissions. Therefore, back trajectories are usually not helpful for explaining OH reactivity over the Suez Canal.

Four case studies (orange numbered labels in Fig. 5) will elucidate the origin of OH reactivity (reported with $\pm$ 1 σ total uncertainty) measured over the Suez Canal:

1) This 4 h period of higher OH reactivity ($15.2 \pm 5.8$ s$^{-1}$ to $19.1 \pm 6.6$ s$^{-1}$) occurred during a waiting period at the southern entrance to the Suez Canal, in the port area of Suez City. Interestingly, OH reactivity was dominated by alkanes here ($\sim$10−15 %). This can be explained by the wind coming from north where directly at the shore a petrol storage area is located (Nasr Petroleum Company, Suez, Egypt). The toluene/benzene ratio was between 1 and 3, indicating fuel combustion emissions as a source in Suez City and/or from ships, rather than biomass burning which tends to be richer in benzene.

2) The Great Bitter Lake is part of the Suez Canal and is used by ships as a "passing lane". During the 4 h waiting time in the lake, the highest OH reactivity of the transit was observed ($20.6 \pm 6.9$ s$^{-1}$ to $26.6 \pm 8.2$ s$^{-1}$). Alkenes, aromatics and OVOCs contributed almost equally to this value ($\sim$16 %, $\sim$17 %, and $\sim$17 %, respectively). Especially the large amount of aromatics is notable. With $7 \pm 1$ % contribution, alkanes were less relevant here than in the Suez City port (case study 1). A visible source of nearby emissions at the northern lakeshore (with northerly winds) was a power plant running on natural gas and/or heavy fuel oil (Abu Sultan Power Station, Ismailia, Egypt (Global Energy Observatory, 2015). Its emissions may have been mixed with those from the lakeshore settlements and from bypassing cargo ships, with a toluene/benzene ratio between 1 and 3, characteristic for fuel combustion emissions.

3) This point of observation was low in OH reactivity ($5.5 \pm 3.2$ s$^{-1}$) as well as in relative humidity (29.2 %). Combined with the back trajectory, this indicates that the air mass sampled here had passed over the Sinai Desert. In this location, the $C_3$ carbonyls/propane ratio was 20.7, indicating a photochemically aged air mass. For comparison, the ratio was never above 10 anywhere else in the area, and in case studies 1 and 2 even between 0 and 1. Similarly, the $C_5$ carbonyls/pentanes ratio was

elevated (2.6) and the toluene/benzene ratio was low (0.5).

4) The wind direction at case study point 4 was north/north-west. Total OH reactivity was in a 4 h peak of $11.8 \pm 5.0$ s$^{-1}$ to $18.2 \pm 6.4$ s$^{-1}$. Ca. 15 % of it was caused by NO$_x$ and ~ 9 % by alkenes. The toluene/benzene ratio was ~1, which indicates a fuel combustion source. There is a highway located west of this northern part of the Canal (Ismailia−Port Said Road), but given the wind direction, exhaust plumes of other vessels sailing in front of the research ship were probably relevant for the

OH reactivity as well as traffic on land.

### 3.3 How many and which chemical species contribute to total OH reactivity around the Arabian Peninsula?

Many studies of ambient total OH reactivity show significant unattributed or "missing" OH reactivity (Yang et al., 2016). Care must be taken in interpreting this missing fraction as it depends on which individual compounds are taken into account,

although forest studies have tended to show greater missing reactivity fractions than urban (Williams and Brune, 2015). Using a total of 120 chemical species (listed in Table S1), we calculated the speciated OH reactivity in 50 min time resolution for the whole AQABA campaign. For the datapoints with speciated OH reactivity above the CRM detection limit, the unattributed fraction of total OH reactivity is plotted against the number of compounds taken into account in Fig. 6. Only datapoints above the detection limit were chosen in order to exclude seemingly unattributed OH reactivity which is actually

due to the detection limit (unattributed fractions for all datapoints are shown along with those for above the LOD in Table 1). The graph shows that when up to ca. 100 species are considered, the unattributed OH reactivity fraction decreases. The remaining compounds collectively contribute only minor additional OH reactivity (less than 0.012 s$^{-1}$). Total OH reactivity is best explained by measured trace gases in the Arabian Gulf ($22 \pm 56$ % unattributed), in the Suez Canal and southern Red Sea ($37 \pm 34$ % and $35 \pm 55$% unattributed, respectively). The largest unattributed OH reactivity was found in the Gulf of

Aden ($55 \pm 38$ %). Other regions with significant unattributed fractions were the northern Red Sea ($49 \pm 39$ %) and the Gulf of Suez ($53 \pm 35$ %). The unattributed reactivity fraction does not correlate with the indicators used for estimating the air mass age.

Figure 7 shows the average OH reactivity of VOCs by molecular mass (NO$_x$ and other inorganic compounds are not considered in this graph) and color-coded for groups of compounds for the Arabian Gulf and the Suez Canal. Higher-mass

aromatics play a larger role in the Suez Canal while the same OVOCs are important in both regions. Generally, less VOCs in the higher mass ranges were measured, which is likely due to larger molecules being partially lost in the ca. 10 m long, 50 °C inlet line. This means that some less volatile components of total OH reactivity might have been missed in this setup. As the

same inlet was used for VOC and OH reactivity measurements, however, this does not impact the unattributed reactivity fraction.

The 10 chemical species which contributed most to OH reactivity in each region traversed during the AQABA campaign are displayed in Fig. 8. Generally, in no regional average was one single compound responsible for an OH reactivity of more than 1.4 s$^{-1}$. This reflects the previous discussion highlighting the necessity of considering a large number of species to get a complete picture of total OH reactivity. The concentrations of some of the compounds among the 10 most relevant ones were derived with the theoretical approach (see Sect. 2.5) and therefore have larger uncertainties. However, their potential importance underlines that a focus on the 14 compounds calibrated with a typical PTR−MS gas standard would not be sufficient for a comprehensive understanding of OH reactivity. Overall, the contribution of the 78 species derived by the theoretical approach amounts to between 22 % (Gulf of Suez) and 51 % (southern Red Sea) of speciated OH reactivity (see Table 1).

The highest percentage contribution to total OH reactivity by the 10 most important compounds was seen in the Gulf of Oman (39 ± 20 %) and the Arabian Gulf (38 ± 20 %), amongst which were the four theoretically calibrated exact masses of acrolein, MBO/pentanal, $C_4$ carbonyls and $C_6$ dienes.

The importance of the seaways around the Arabian Peninsula for maritime transport is reflected in the large contribution of $NO_2$ and CO (associated with fuel combustion), which are among the three highest contributors to OH reactivity in most of the regions and among the top 10 in all of them. Oxygenated species such as aldehydes and ketones were important OH sinks in all the areas, in particular formaldehyde, acetaldehyde and, to a lesser extent, the $C_4$ and $C_5$ carbonyls. Directly emitted hydrocarbons were among the 10 most important OH reactivity contributors only in areas closer to emission sources: e.g. n-butane and propane in the Gulf of Suez, or unsaturated hydrocarbons such as $C_6$ dienes, $C_5$ dienes (without isoprene) and ethene in the Suez Canal. The importance of oxygenates in the air composition during AQABA reflects the fast photooxidation of direct emissions during this summertime campaign (also see Sect. 3.1.1). In winter, the directly emitted hydrocarbons would have longer lifetimes and therefore gain importance in the OH reactivity budget at the expense of oxygenates.

In the Arabian Sea, which was the area with the lowest total OH reactivities (Fig.1, Table 1), the global background OH reactivity from methane was a major OH sink. Dimethyl sulfide (DMS), a biogenic molecule emitted by algae which has been observed in this part of the ocean before (Warneke and Gouw, 2001) and will be discussed elsewhere for the AQABA campaign (Edtbauer et al., 2019, in preparation), appears among the 10 most important OH sinks here despite a reactivity of only 0.02 s$^{-1}$. This is evidence for relatively clean air due to the influence of open ocean air, although even here emissions from shipping are evident from the CO and $NO_2$ contributions. In the Mediterranean, methane was the second-most important single contributor to total OH reactivity after CO.

Generally, anthropogenic VOCs and their oxidation products dominated OH reactivity around the Arabian Peninsula. For example, acrolein was among the 10 most important contributors almost everywhere during AQABA (Fig. 8). Acrolein is a major product of the oxidation of 1,3-butadiene, which is among the most abundant alkenes emitted from anthropogenic

sources (Grosjean et al., 1994). The $C_4$ carbonyls (butanal or methyl ethyl ketone (MEK), are major contributors to total OH reactivity over the Arabian Basin, and both can be directly emitted from fossil fuel combustion (Schauer et al., 2002) or be the result of ambient oxidation processes. Carbonyl photolysis has been found recently to be a possible source of ozone formation in an oil and gas production area (Edwards et al., 2014).

### 3.4 OH reactivity and ozone

### 3.4.1 Ozone production regimes

Ozone and OH reactivity are connected via a chemical cycle that involves $NO_x$ and VOCs (Sillman, 1999; Ren et al., 2013): OH oxidizes VOCs to form peroxy radicals, which, in turn, oxidize NO to $NO_2$. $NO_2$ thus formed is subsequently photolyzed to result in ozone production. Ozone formation is, however, neither linearly dependent on $NO_x$ concentration nor VOC reactivity, and a decrease in one parameter does not necessarily lead to a decrease in ozone formation (Pusede and Cohen, 2012).

Defining ozone production regimes in terms of the OH reactivities of VOCs and $NO_x$ is a way of assessing the sensitivity of ozone production to the prevailing conditions (Kirchner et al., 2001; Sinha et al., 2012). The method and underlying chemistry has been extensively evaluated for different conditions and compared to other methods in Kirchner et al. (2001). The amount of peroxy radicals produced by VOC oxidation is linked to the OH reactivity of VOCs. The OH reactivity of VOCs was calculated here as the difference of measured total OH reactivity (in 5 min time resolution) and OH reactivity of measured $NO_x$. Only data at daytime (06:00–18:00 local time) was considered, because the method is inherently limited to daytime chemistry. Figure 9 shows ozone production regime plot examples, where "s" denotes the relative reactivity of OH towards $NO_x$ and VOCs. Based on the analysis of Kirchner et al. (2001), which was developed for urban VOC mixtures, in between the lines labelled s = 0.2 and s = 0.01, ozone production is favored by a suitable ratio of $NO_x$ and OH reactivity of VOCs, whereas s > 0.2 indicates VOC limitation, and s < 0.01 $NO_x$ limitation. The Arabian Sea with its comparably clean air was a region with partially $NO_x$ limited ozone production (Fig. 9a). Here, 20 % of the datapoints do not fall within the regime of ozone production, mostly due to $NO_x$ limitation. This is reflected in relatively low ozone mixing ratios. The Gulf of Suez (Fig. 9b) is characterized in part by a strong VOC limitation associated with high $NO_x$ (69 % of datapoints indicate an ozone formation regime). This $NO_x$ can be attributed to emissions from other vessels at close proximity while waiting at the entrance of the Suez Canal. The low ozone mixing ratios associated with high $NO_x$ in the fresh combustion emissions (Fig. 9b/c) point to ozone titration, which will be discussed in greater detail in Sect. 3.4.2. In the Suez Canal (Fig. 9c), the datapoints are also grouped more towards VOC sensitivity whereas the opposite is true for the Arabian Gulf (Fig. 9d). Nevertheless, in both the Suez Canal and the Arabian Gulf, almost all datapoints (97 % in the Arabian Gulf and 91 % in the Suez Canal) fall into the regime of ozone formation, meaning that a suitable ratio of VOC and $NO_x$ molecules for ozone formation was present nearly all the time. The same is true for the other environments (not shown in graph) with favored

ozone formation 97 % of the time in the Mediterranean, 93 % in the northern Red Sea, 88 % in the southern Red Sea, 87 % in the Gulf of Aden, and 82 % in the Gulf of Oman.

The ozone production favoring conditions with both high $NO_x$- and VOC-attributed OH reactivities may be contributing factors to the high ozone levels observed and modeled above the Arabian Gulf (Lelieveld et al., 2009; Fountoukis et al., 2018). Ozone formation favorable conditions resulting in ozone mixing ratios up to 145 ppb have been reported from other sites influenced by oil and gas production (Edwards et al., 2014; Wei et al., 2014). Moradzadeh et al. (2019) found that a hydrocarbon processing plant at the shore of the Arabian Gulf impacted ozone levels even in 300 km distance. Generally, emissions in lower latitudes (e.g. below 30° N as in the Arabian Gulf) lead to more efficient formation of tropospheric ozone than in higher latitudes, because the meteorological conditions lead to faster reaction rates and strong convection (Zhang et al., 2016).

### 3.4.2 OH reactivity/ozone correlations

The relationship between ozone and total OH reactivity during the AQABA campaign is depicted in Fig. 10. Data from most regions display a negative correlation between ozone mixing ratio and total OH reactivity. This is exemplified for the Mediterranean, the Suez Canal and Gulf of Suez in Fig. 10a (correlation for Suez Canal: slope = -0.2; $r^2 = 0.61$). Additionally, there is a large group of datapoints located at lower OH reactivity (below 10 $s^{-1}$) and ozone mixing ratios of 50 to 70 ppb.

The reason for the negative correlation seen in parts of the data is displayed in Fig. 10b. Higher OH reactivities in this part of the plot coincide with higher $NO_x$, stemming from the exhaust plumes of vessels passing close by. In these ship plumes, high levels of VOCs are co-emitted with $NO_x$. This combination leads to elevated total OH reactivity. At the same time, ozone is depleted by the NO emissions in the fresh ship plumes ($NO + O_3 \rightarrow NO_2 + O_2$, Sillman, 1999). This effect is seen when an emission source co-emitting $NO_x$ and VOCs is very close (Akimoto, 2016), as ozone values recover rapidly downwind through $NO_2$ photolysis and mixing. The ozone titration effect has similarly been observed in negative ozone-CO correlations (Parrish et al., 1998).

The bulk of datapoints between 50 and 70 ppb of ozone associated with lower OH reactivity and no trend is, on the other hand, more representative of a campaign background/average, uninfluenced by immediate emission sources and with lower $NO_x$ (Fig. 10b). Here, a possible longer-term effect of the co-emission of VOC and $NO_x$ from marine transport and other anthropogenic sources can be seen. As a secondary pollutant and being suppressed by NO, ozone can often be elevated at distance to the pollution sources (Sudo and Akimoto, 2007). Large quantities of ozone can be produced during long-range transport under sunlight influence (Parrish et al., 1998; Akimoto, 2016; Moradzadeh et al., 2019). Consequently, in the seaways around the Arabian Peninsula, ozone mixing ratios were mostly between 50 and 80 ppb, which is the range of daytime values in polluted Beijing (Williams et al., 2016; Yang et al., 2017).

Over the Arabian Gulf, the same negative relationship between OH reactivity and ozone as in the Suez Canal, Gulf of Suez and Mediterranean existed in those datapoints where ozone was below 70 ppb. In contrast, at ozone mixing ratios above ca.

80 ppb, no trend or a slightly positive correlation between ozone mixing ratio and OH reactivity can be ascertained (Fig. 10a). Ozone mixing ratios above 70 ppb over the Arabian Gulf were associated with a high fraction of OVOCs (Fig. 10c), indicating more photochemically processed air. Additionally, when OH reactivity was high at the same time as ozone, alkene OH reactivity was also elevated (Fig. 10d). This points to a mixing in of fresh emissions (associated with short-lived alkenes) into photochemically aged, oxidized air contaminants (high OVOCs and high ozone) over the Arabian Gulf. Polluted air masses from urban or industrial areas on land were probably oxidized during their transport to the coast/sea, where they were mixed with emissions from oil and gas extraction facilities, as has been discussed in detailed case studies of such occasions in Sect. 3.2.

## 4 Summary and conclusions

In July and August 2017, during the AQABA campaign, the first ship-based direct measurements of total OH reactivity were conducted using the Comparative Reactivity Method. The total OH reactivity in ambient air around the Arabian Peninsula was predominantly related to anthropogenic influence. Particularly, reactive hydrocarbons from oil and gas extraction/production and inorganic reactants such as $NO_x$ and CO from fuel combustion by shipping were major OH sinks during the whole campaign. Total OH reactivity over the Arabian Basin varied, with periods below the detection limit over the Arabian Sea, regional medians of 7.9−8.5 s$^{-1}$ under permanent maritime traffic influence over the seaways of the Red Sea, Gulf of Aden and Gulf of Oman, and episodes of up to 32.8 $\pm$ 9.6 s$^{-1}$ when air masses were influenced by emissions from oil/gas extraction facilities, urban centers and/or ships in close proximity. The largest regional median OH loss rate of 18.8s$^{-1}$ was observed over the Arabian Gulf when air originated from Iraq/Kuwait, where large oil fields and refineries at the shore provided high emissions. Over the Suez Canal, median OH reactivities of 10.8 s$^{-1}$ and up to 26.6 $\pm$ 8.2 s$^{-1}$ were attributed to local emissions from a petroleum storage facility, a power plant and traffic on sea and land. The cleanest environment was the Arabian Sea, where the marine biogenic compound dimethyl sulfide was among the 10 most important known contributors to OH reactivity.

In all the regions around the Arabian Peninsula, OVOCs were the class of compounds that provided the largest identified OH sink. They were formed efficiently because photochemistry was highly active due to intense solar irradiation and high temperatures in summer. Photochemically aged air mixed with fresh nearby emissions over the Arabian Gulf, leading to extremely high ozone mixing ratios. Using ozone production regime plots, we found that the ratio of VOC- and $NO_x$-attributed OH reactivities was favorable for ozone formation nearly all along the ship track. In the Arabian Gulf, large OH reactivity from VOCs due to oil and gas sources together with $NO_x$ from marine traffic lead to a regime of ozone production 97 % of the time. The only regions where ozone production was limited for a significant fraction of time were the Gulf of Suez with VOC limitation when affected by $NO_x$-rich plumes, and the Arabian Sea with a slight $NO_x$ limitation due to clean air from the open Indian Ocean. Ozone versus OH reactivity correlation plots were found to be a valuable tool in identifying

different chemical regimes with regards to ozone formation and loss: ozone destruction in $NO_x$-rich ship plumes was clearly distinguishable from a mix of fresh petroleum extraction emissions with photochemically aged air.

Studies of ambient total OH reactivity often show a significant "missing" or unattributed fraction of OH reactivity, although this tends to be smaller in anthropogenic environments (Williams and Brune, 2015). In an attempt to close the OH reactivity

budget, the OH reactivity of compounds with unidentified structure from PTR−ToF−MS measurements was calculated using averages of the reaction rate constants of all possible structures attributed to each chemical formula. These species were important OH sinks, accounting for 22−51 % of the regional speciated OH reactivity. With this approach, the measured total OH reactivity can be explained within the uncertainty for the Suez Canal, the southern Red Sea, the Gulf of Oman and the Arabian Gulf. Significant unattributed fractions remain in the Gulf of Suez, the northern Red Sea and the Gulf of Aden.

Overall, ~ 100 chemical species needed to be identified to explain the measured total OH reactivity within the uncertainty, while the 10 most important compounds contributed 20−39 % of the reactivity. Unattributed reactivity in plumes from hydrocarbon processing facilities may be due to branched and/or higher hydrocarbons not captured with the techniques available on board the *Kommandor Iona*. In order to get a more comprehensive picture of the chemical species relevant for OH chemistry in the region, measurements with even greater chemical detail in the hydrocarbons would be necessary.

The range and composition of OH reactivity (and the ozone mixing ratios) around the Arabian Peninsula were, despite the marine measurement locations, broadly comparable to observations from highly populated urban areas, except for an unusually large contribution of OVOCs (Kovacs et al., 2003; Mao et al., 2010; Dolgorouky et al., 2012; Kim et al., 2016; Williams et al., 2016; Praplan et al., 2017; Yang et al., 2017). The "urban-like" OH reactivity is related to intensive international ship traffic in the seaways in the Arabian Basin and to major emissions from oil and gas industries on sea

platforms and at the coasts. Oil and gas related VOCs were important sources of OH reactivity, and could be a relevant factor for the extremely high ozone concentrations and the photochemical air pollution usually seen in summer in the region (Lelieveld et al., 2009; Smoydzin et al., 2012; Farahat, 2016; Barkley et al., 2017).

Most predictions assume that oil and natural gas production will increase in the Middle East during the coming decades (Balat, 2006; Khatib, 2014; Holz et al., 2015; Overland, 2015). Similarly, ship emissions around the Arabian Peninsula have

been projected to grow on the short term due to intensifying traffic (Eyring et al., 2007), although new emission regulations set by the International Maritime Organization (IMO, 2019), effective from 2020, might lead to a long-term decrease. Given that the conditions favor ozone production and will increase to do so due to rising temperatures (Wasimi, 2010; Lelieveld et al., 2012; Lelieveld et al., 2016), emission regulations in the region appear to be the only way to prevent a further increase in photochemical air pollution in the future.

**Data availability**

The data used in this study will be available from August 2019 to all scientists agreeing to the AQABA protocol at https://doi.org/10.5281/zenodo.3354117.

## Author contributions

EP and NW were responsible for OH reactivity measurements and data. EP conducted further data analysis and wrote the first draft of the manuscript. AE, EB, LE and CS contributed VOC and NMHC measurements and data. Ozone and $SO_2$ data were provided by JC and PE, formaldehyde and $NO_x$ data by DD, BH, IT and HF, methane and carbon monoxide by JP. DW calculated the back trajectories. JL designed and realized the campaign. JW supervised the study. All authors contributed to manuscript writing and revision, read and approved the submitted version.

## Competing interests

The authors declare that they have no conflict of interest.

## Acknowledgements

We thankfully acknowledge the cooperation with the Cyprus Institute (CyI), the King Abdullah University of Science and Technology (KAUST) and the Kuwait Institute for Scientific Research (KISR). We thank Hays Ships Ltd, Captain Pavel Kirzner and the ship crew for their support on-board the *Kommandor Iona*. We would like to express our gratitude to the whole AQABA team, particularly Hartwig Harder for daily management of the campaign; and Marcel Dorf, Claus Koeppel, Thomas Klüpfel and Rolf Hofmann for logistical organization and help with preparation and setup. We are grateful for Uwe Parchatka's assistance with $NO_2$ loss characterizations and Jan Schuladen's solar radiance data. The campaign was funded by the Max Planck Society. The position of Nijing Wang was funded by the European Union's Horizon 2020 research and innovation programme under the Marie Skłodowska-Curie grant agreement No. 674911.

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

**Table 1: Overview of measured and speciated OH reactivity by region. "≥ LOD" signifies the use of exclusively those datapoints where speciated OH reactivity is equal to or above the LOD. Percentages of contribution to total OH reactivity by group were calculated from all datapoints in 50 min time resolution. The chemical species attributed to each group are listed in Table S1. For an explanation of unattributed or speciated OH reactivity and "theoretical approach" species refer to Sect. 2.5. The 10 most important species are the species with largest average OH reactivity over the respective region. The ratios of $C_3$ or $C_5$ carbonyls vs. their precursors or toluene vs. benzene are indicators for air mass age. *Note that the detection limit was 5.4 $s^{-1}$. (Medit. = Mediterranean.)**

| | Medit. Sea | Suez Canal | Gulf of Suez | Red Sea north | Red Sea south | Gulf of Aden | Arabian Sea | Gulf of Oman | Arabian Gulf |
|---|---|---|---|---|---|---|---|---|---|
| **Aromatic hydrocarbons** | 1 % | 10 % | 3 % | 4 % | 8 % | 3 % | 2 % | 5 % | 7 % |
| **Alkanes** | 4 % | 5 % | 6 % | 5 % | 4 % | 4 % | 5 % | 4 % | 9 % |
| **Alkenes** | 2 % | 10 % | 4 % | 6 % | 9 % | 4 % | 3 % | 5 % | 9 % |
| **OVOCs** | 11 % | 14 % | 9 % | 14 % | 14 % | 12 % | 12 % | 21 % | 35 % |
| **NVOCs** | 1 % | 1 % | 0 % | 1 % | 1 % | 1 % | 1 % | 1 % | 1 % |
| **SVOCs** | 0 % | 0 % | 0 % | 0 % | 0 % | 0 % | 1 % | 1 % | 0 % |
| **Halogenated VOCs** | 0 % | 0 % | 0 % | 0 % | 0 % | 0 % | 0 % | 0 % | 0 % |
| **Inorganics** | 8 % | 6 % | 5 % | 6 % | 5 % | 4 % | 4 % | 5 % | 6 % |
| **$NO_x$** | 4 % | 10 % | 12 % | 5 % | 5 % | 4 % | 1 % | 9 % | 5 % |
| **Unattributed OH reactivity fraction (all datapoints)** ± 1 σ total uncertainty | 68 ± 52 % | 44 ± 43 % | 59 ± 44 % | 59 ± 50 % | 54 ± 54 % | 68 ± 51 % | 72 ± 57 % | 49 ± 55 % | 27 ± 55 % |
| **Unattributed OH reactivity fraction ≥ LOD** ± 1 σ total uncertainty | none | 37 ± 35 % | 53 ± 34 % | 49 ± 39 % | 35 ± 55 % | 55 ± 38 % | none | 41 ± 52 % | 22 ± 56 % |
| **Fraction of species with concentrations derived from theoretical approach** | 28 % | 28 % | 22 % | 38 % | 51 % | 46 % | 42 % | 46 % | 46 % |
| **Speciated (calculated) OH reactivity [$s^{-1}$], averages** ± standard deviation | **2.3** ± 1.2 | **7.4** ± 5.8 | **4.7** ± 2.2 | **3.7** ± 1.7 | **4.0** ± 2.1 | **2.6** ± 1.1 | **1.5** ± 0.2 | **4.3** ± 1.2 | **9.3** ± 5.6 |
| **Speciated (calculated) OH reactivity [$s^{-1}$], medians,** 25th−75th percentile | **2.0** 1.9−2.2 | **5.4** 2.7−13.4 | **4.3** 3.0−5.9 | **3.3** 2.5−4.5 | **3.7** 2.2−5.5 | **2.4** 2.0−2.9 | **1.5** 1.4−1.7 | **4.0** 3.2 −5.2 | **7.6** 5.2 −11.9 |
| **Measured OH reactivity [$s^{-1}$], averages** ± standard deviation | **7.2** ± 2.9 | **13.2** ± 6.9 | **11.6** ± 4.2 | **9.1** ± 3.3 | **8.7** ± 4.2 | **8.2** ± 3.1 | **5.6*** ± 2.0 | **8.3** ± 3.6 | **12.9** ± 6.2 |
| **Measured OH reactivity [$s^{-1}$], medians,** 25th−75th percentile | **6.8** 5.8−7.8 | **10.8** 7.4−18.8 | **10.4** 8.2−15.0 | **8.5** 7.0−10.5 | **7.9** 5.7−10.9 | **8.0** 5.9−10.1 | **4.9*** 4.1−6.5 | **8.4** 5.9−10.5 | **11.2** 8.9−15.2 |
| **OH reactivity of 10 most important species [$s^{-1}$]** | 1.7 | 3.8 | 3.1 | 2.1 | 2.2 | 1.7 | 1.1 | 3.3 | 4.9 |
| **Fraction of total OH reactivity contributed by 10 most important species** | 24 % | 29 % | 27 % | 23 % | 25 % | 21 % | 20 % | 39 % | 38 % |
| **$C_3$ carbonyls / propane [ppb ppb$^{-1}$], medians** | 8.6 | 3.4 | 2.6 | 1.9 | 9.4 | 6.8 | none | 2.4 | 0.9 |
| **$C_5$ carbonyls / pentanes [ppb ppb$^{-1}$], medians** | 3.0 | 1.7 | 1.1 | 1.3 | 1.9 | 2.4 | 3.8 | 1.2 | 2.8 |
| **Toluene / benzene [ppb ppb$^{-1}$], medians** | 0.24 | 0.88 | 0.80 | 0.39 | 0.78 | 0.33 | 0.18 | 0.79 | 0.47 |

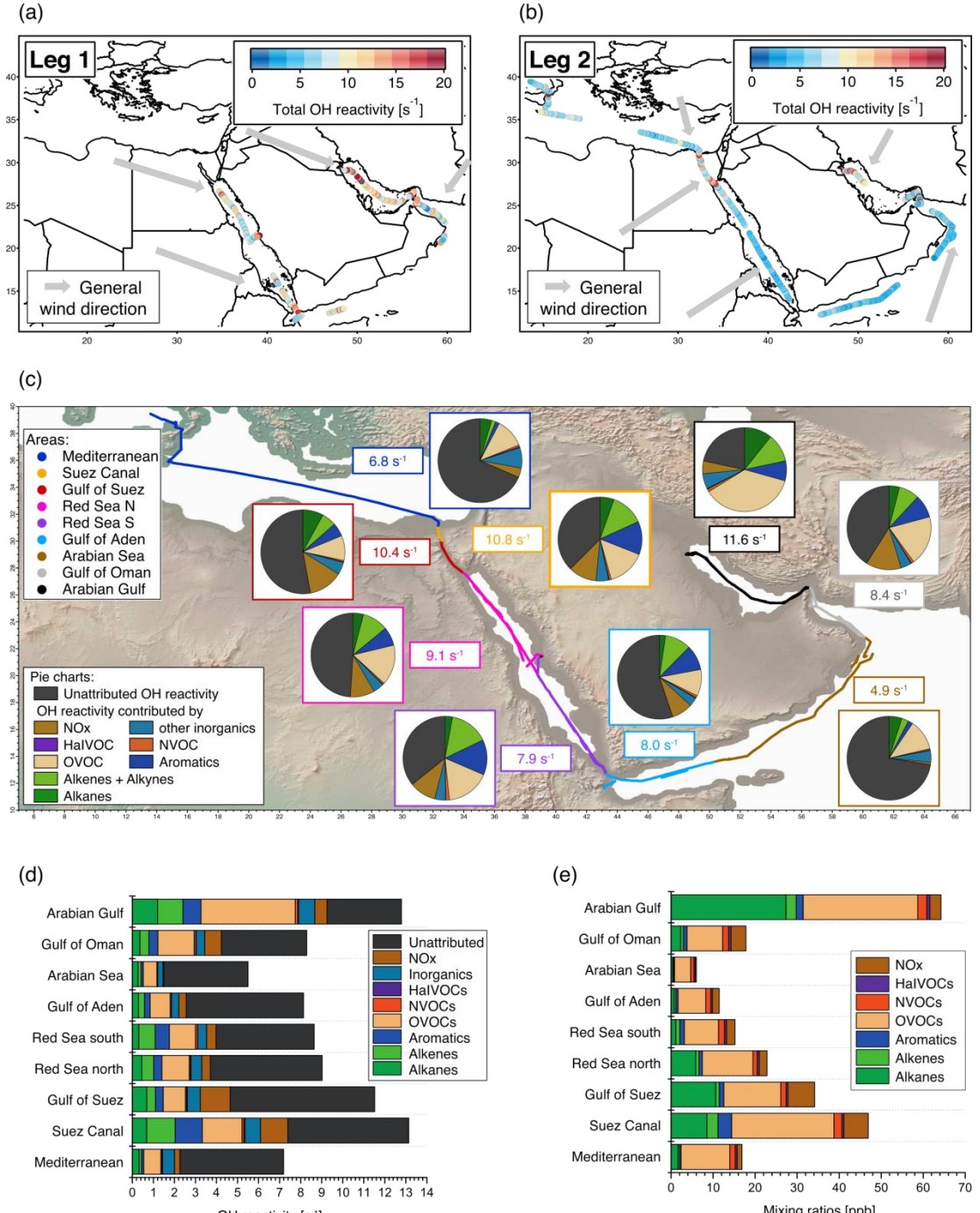

**Figure 1: Overview of total OH reactivity around the Arabian Peninsula during the AQABA campaign.** OH reactivity during (a) leg 1 (July 5−July 31, 2017) and (b) leg 2 (August 3−August 31, 2017). The maximum in the color scales is set to 20 s⁻¹ for better visibility of differences, although there are a few datapoints above this value. Arrows depict general wind directions for the respective regions. (c) Total OH reactivity medians by region, and pie charts showing the contribution of compound classes for

datapoints where speciated OH reactivity ≥ LOD (exception: pie charts of Mediterranean and Arabian Seas show average of all datapoints, due to low number of points above LOD). (d) Average OH reactivity and speciation by region for all datapoints, including those where speciated OH reactivity was below the LOD. Error bars show the total uncertainty of the measurement. (e) Average mixing ratio of VOCs/trace gases in ppb by compound class and region (except for the class of inorganic compounds other than $NO_x$). The individual compounds included in each class are listed in Table S1. Port calls and bunkering are excluded from all averages.

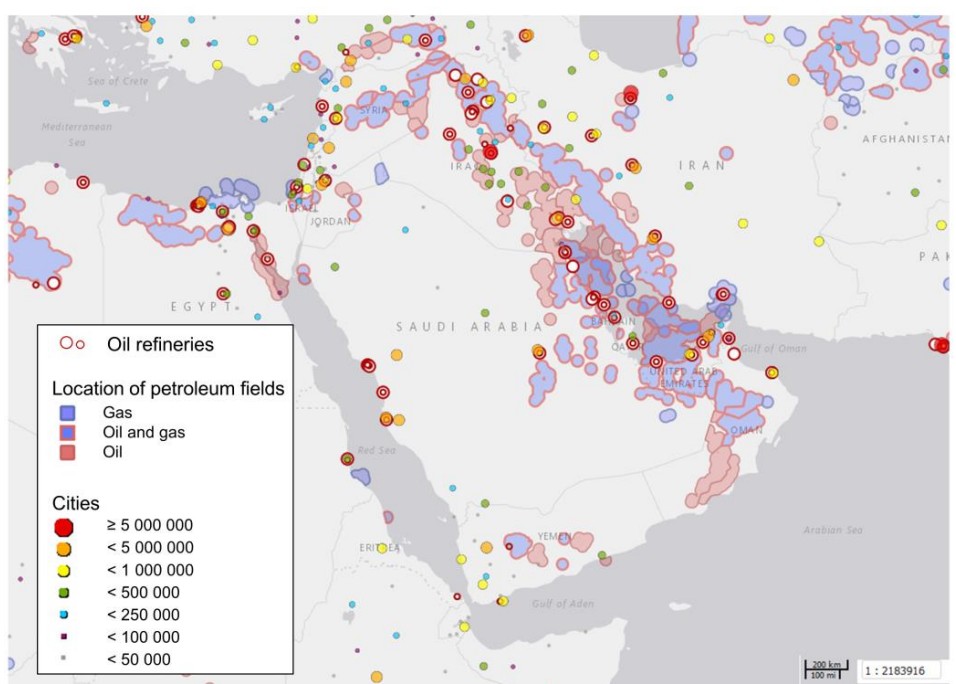

**Figure 2: Oil refineries, petroleum fields and cities in the Arabian Basin. City markers are scaled with the number of inhabitants as denoted in the legend. Source: Harvard WorldMap Project (https://worldmap.harvard.edu/maps/6718/dJT).**

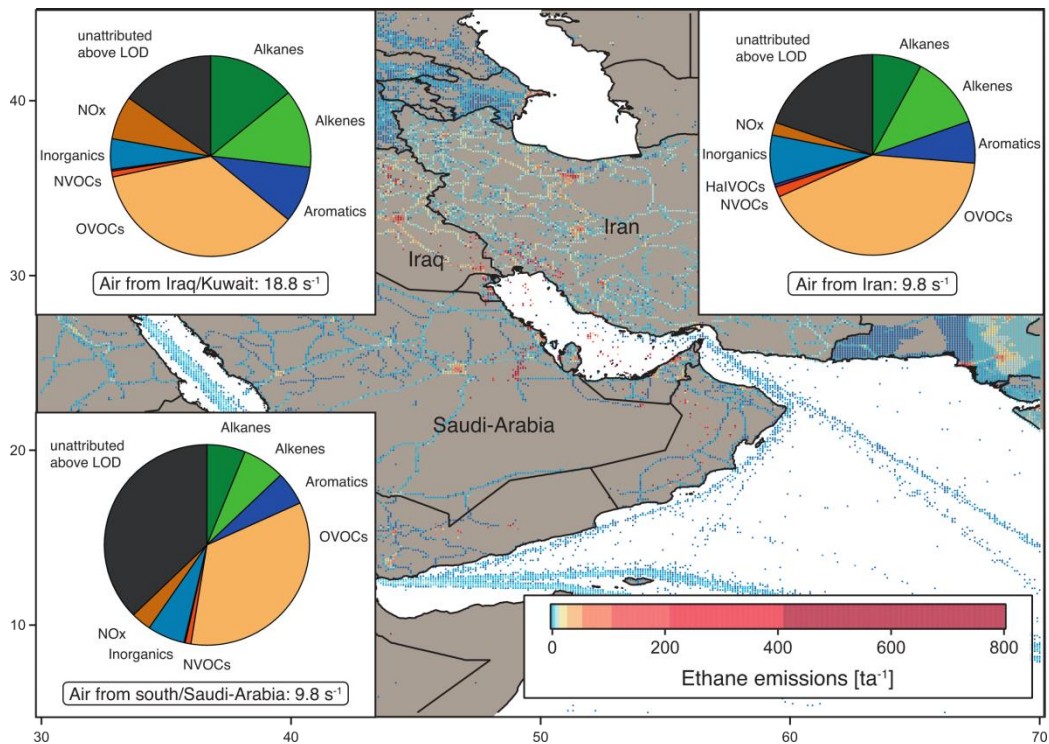

**Figure 3: OH reactivity composition and average ± standard deviation in the Arabian Gulf, separated by origin of the measured air masses: from Iraq/Kuwait, from Iran from southern Arabian Gulf/Saudi Arabia. Background: Ethane emissions map using data from the EDGAR database (http://edgar.jrc.ec.europa.eu, for 2012) as an indicator for the distribution of petroleum extraction and other anthropogenic sources.**

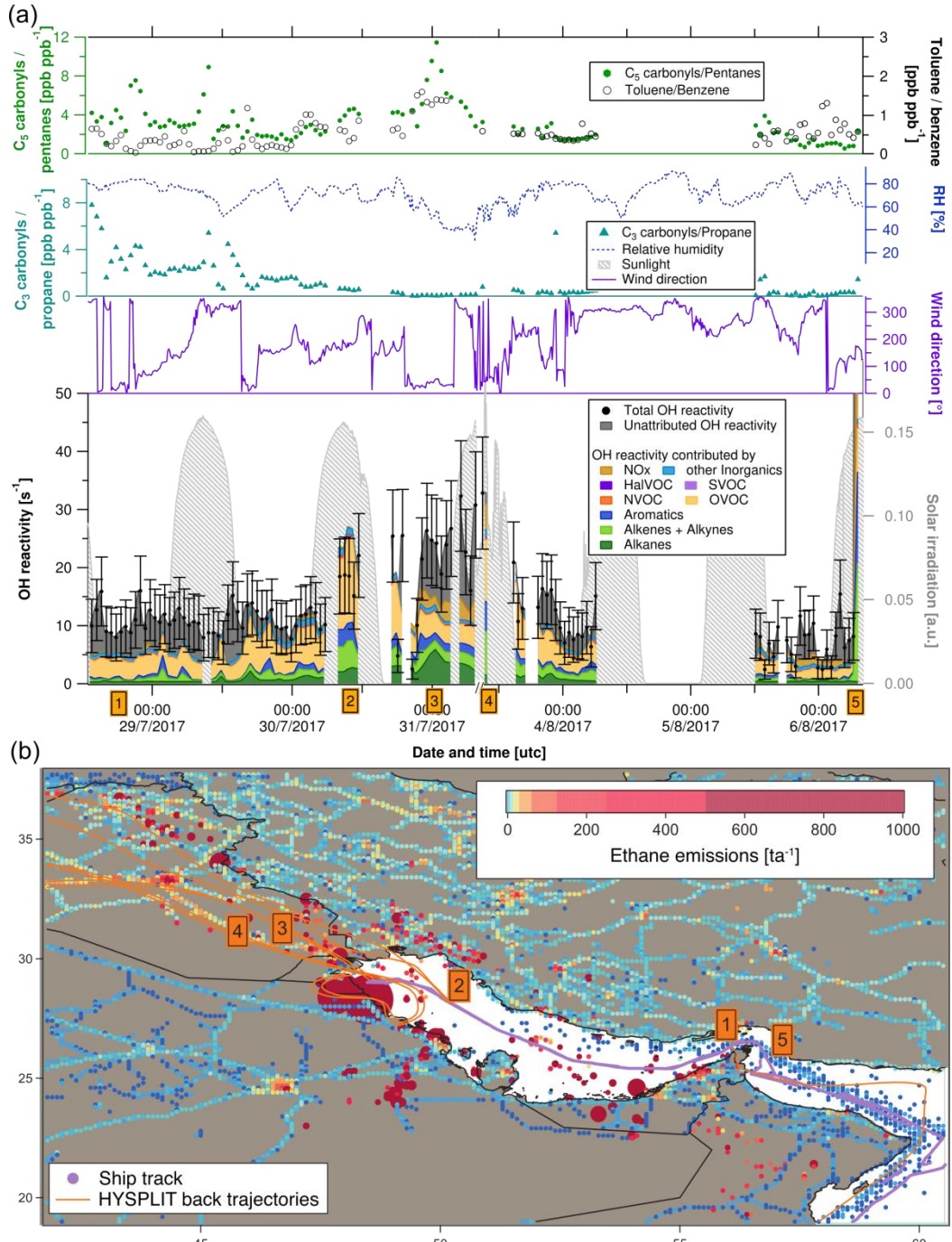

**Figure 4: (a) $C_3$ carbonyls/propane ratio, $C_5$ carbonyls/pentane ratio, toluene/benzene ratio, relative humidity, solar irradiation, wind direction, speciated and measured OH reactivity in the Arabian Gulf. Error bars of total OH reactivity display the 1 σ total uncertainty of the measurement. (b) Ship track of the campaign in the Arabian Gulf with HYSPLIT air mass back trajectories and numbered labels of case study points. Ethane emissions from the EDGAR database (http://edgar.jrc.ec.europa.eu, for 2012) as an indicator for petroleum extraction/processing and other anthropogenic sources are shown on a logarithmic colour scale. Emissions out of scale (> 1000 t $a^{-1}$) are shown with larger circles.**

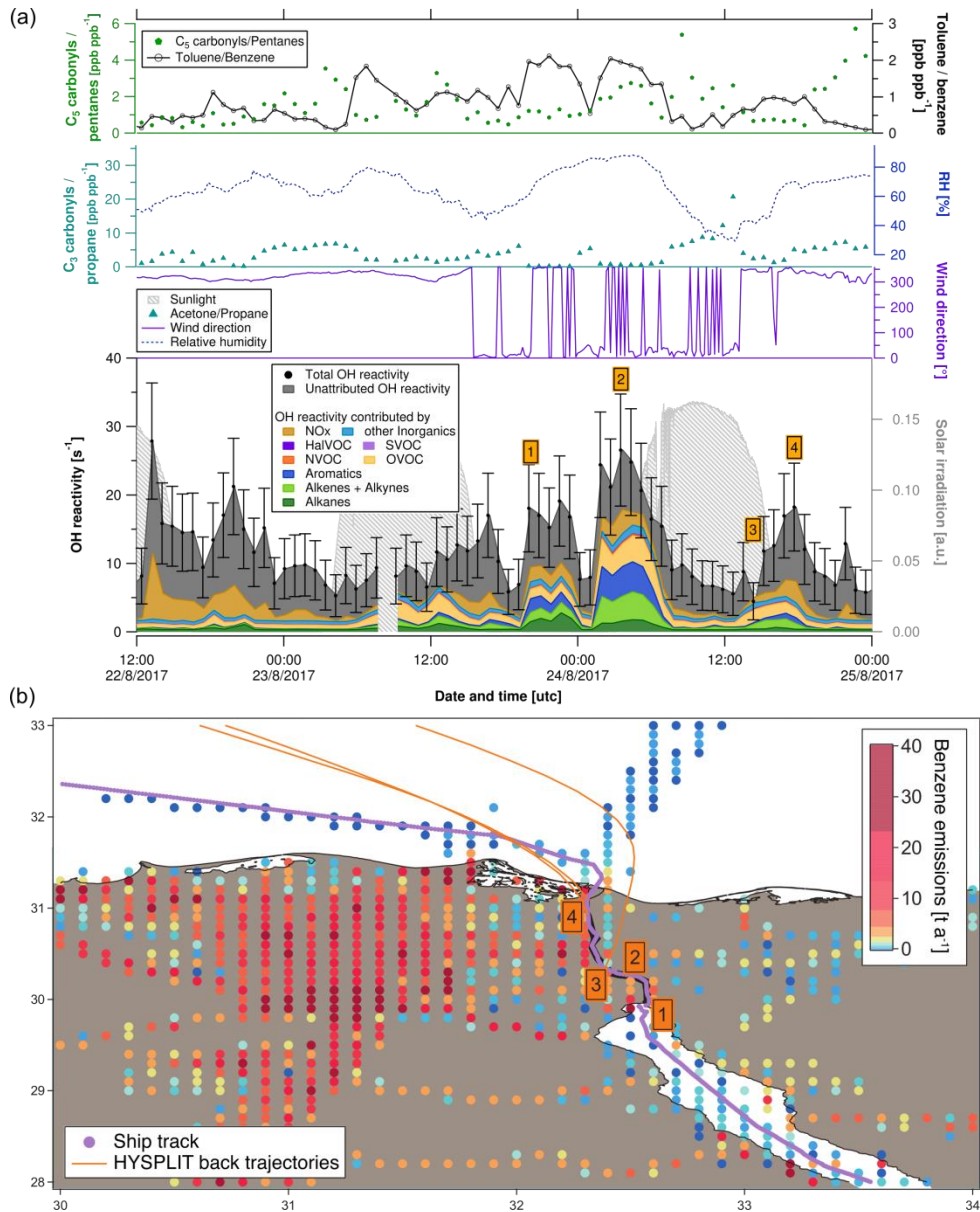

**Figure 5: (a) C$_3$ carbonyls/propane ratio, C$_5$ carbonyls/pentane ratio, toluene/benzene ratio, relative humidity, solar irradiation, wind direction, speciated and measured OH reactivity in the Suez Canal. Error bars of total OH reactivity display the total uncertainty of the measurement. (b) Ship track of the cruise in the Suez Canal with HYSPLIT air mass back trajectories and numbered labels of case study points. Benzene emissions as an indicator for anthropogenic sources from the EDGAR database (http://edgar.jrc.ec.europa.eu, for 2012) are shown on a logarithmic colour scale.**

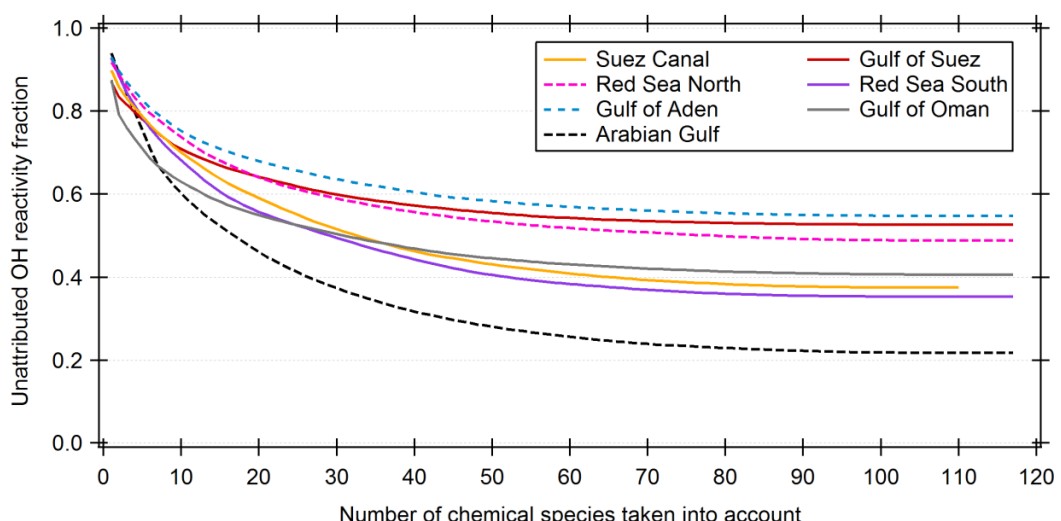

**Figure 6: Unattributed OH reactivity fraction as a function of the number of compounds/chemical species taken into account for calculating speciated OH reactivity. The chemical species are taken into account with decreasing average OH reactivity. Only datapoints where the summed speciated OH reactivity is larger than or equal to the detection limit (LOD) of the CRM measurement are considered here. The Mediterranean and Arabian Sea are not shown because of insignificant number of datapoints above the LOD.**

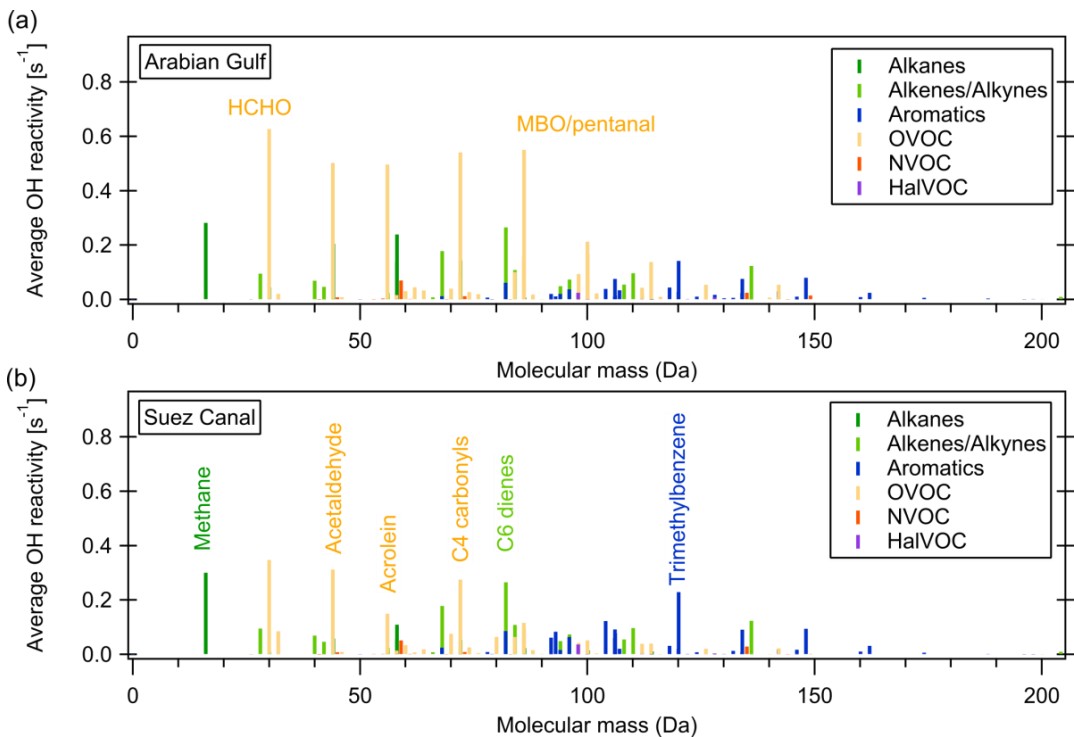

**Figure 7: Average OH reactivity of VOCs by molecular mass in the Suez Canal and Arabian Gulf. Colors show the compound class.**

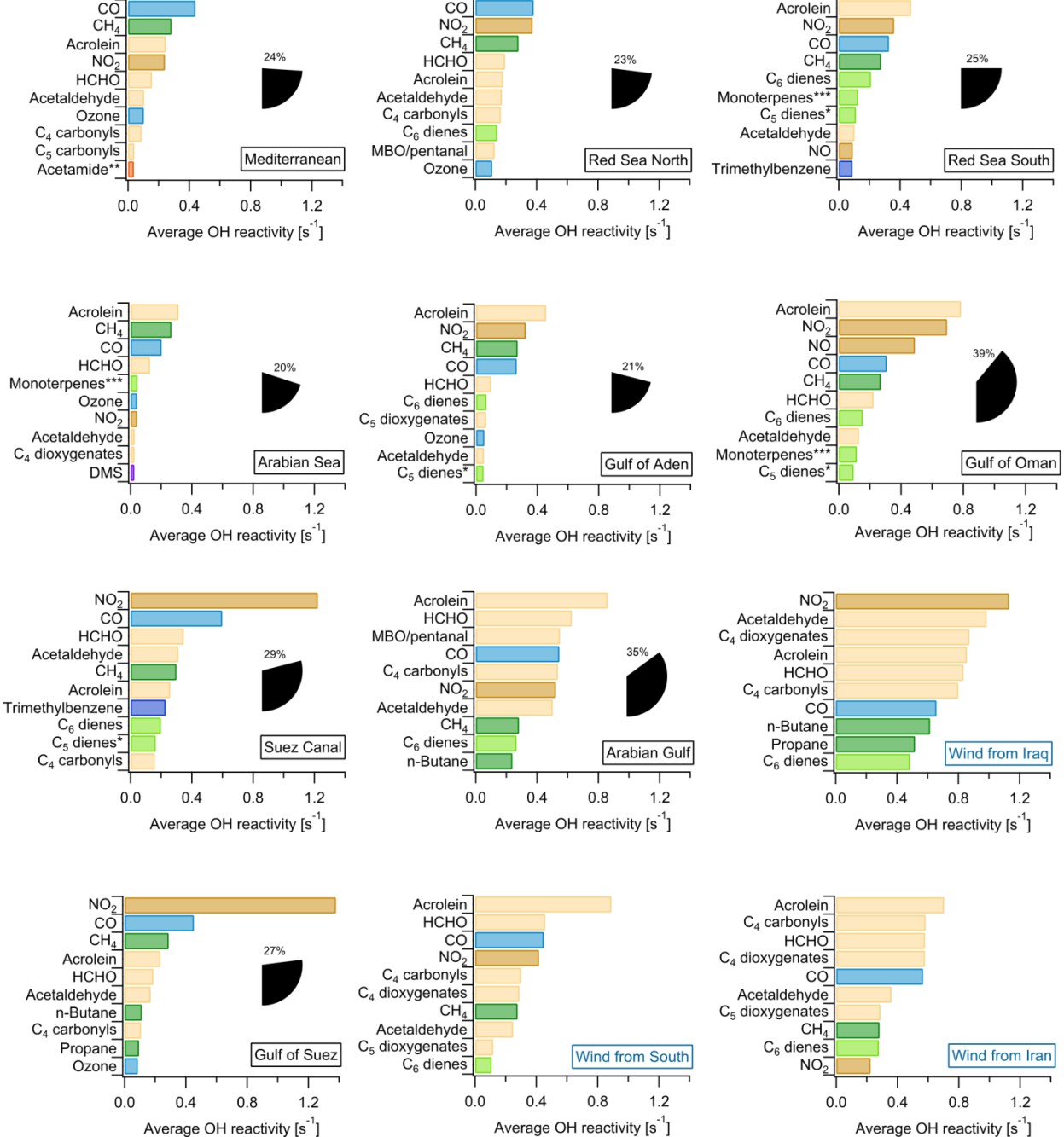

**Figure 8: Regional average OH reactivity of each of the 10 most important contributors to OH reactivity. The percentage of total OH reactivity from these 10 compounds is shown as pie wedges. The Arabian Gulf was split by air mass origin (plots with blue labels). *$C_5$ dienes do not include isoprene. **Acetamide or *N*-methyl formamide. ***"Monoterpenes" may include anthropogenic $C_{10}$ trienes.** Abbreviations: MBO: 2-Methyl-3-buten-2-ol.

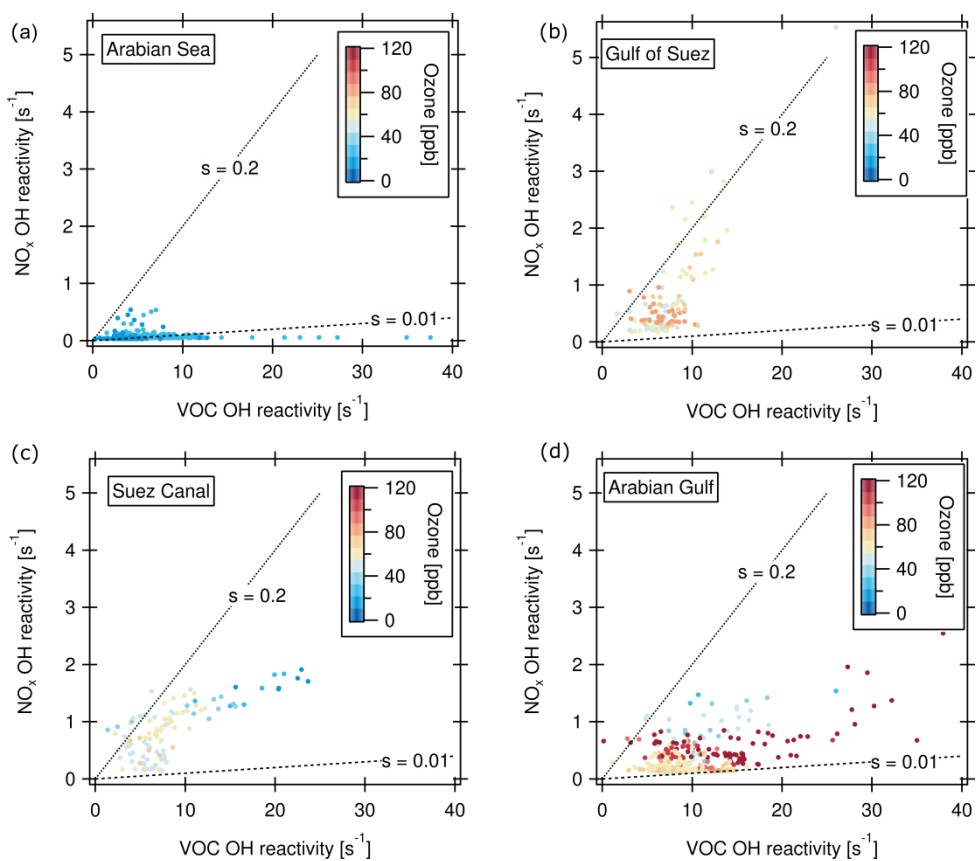

**Figure 9: Ozone production regimes for (a) Arabian Sea, (b) Gulf of Suez (c) Suez Canal, and (d) Arabian Gulf. "s" denotes the relative reactivity of OH towards $NO_x$ and VOCs. For s > 0.2: VOC limitation, for s < 0.01: $NO_x$ limitation of the ozone formation.**

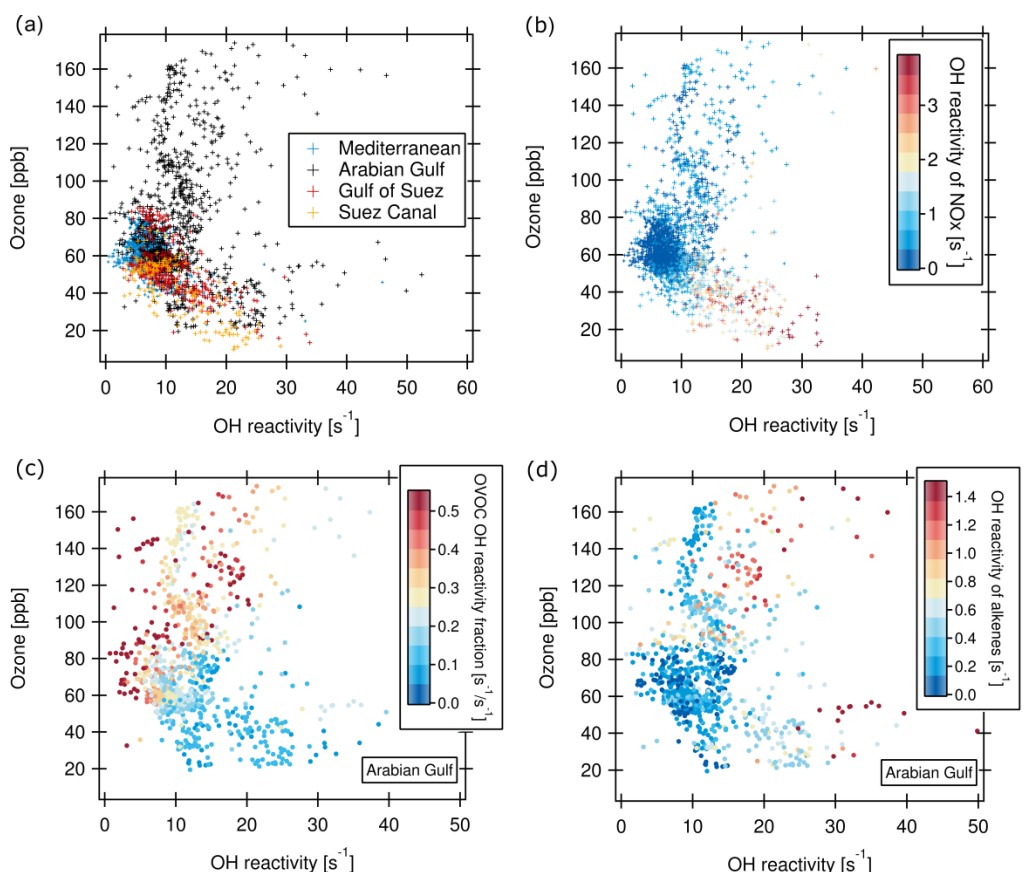

**Figure 10: (a)** Ozone vs. total OH reactivity for four regions. **(b)** Same as (a), but with OH reactivity of $NO_x$ as color scale (no datapoints when $NO_x$ values are missing). **(c)** Ozone vs. OH reactivity for the Arabian Gulf with the fraction of OVOC in total OH reactivity as color scale. **(d)** Ozone vs. OH reactivity for the Arabian Gulf with OH reactivity of alkenes as color scale.

