# Peer review of "Shipborne measurements of total OH reactivity around the Arabian Peninsula and its role in ozone chemistry"

_Atmospheric Chemistry and Physics, 2019_

## Referee Comment (RC1) · Anonymous Referee #1 · 21 Jun 2019

The manuscript describes the measurements of OH reactivity made using the comparative reactivity method (CRM) on a shipborne platform during the Air Quality and Climate Change in the Arabian Basin (AQABA) campaign in 2017. Detailed analysis of the measurements is provided, using supporting measurements of VOCs and other trace gases to determine the sources of reactivity and the contributions to the total OH reactivity from different classes of compounds. Measurements of atmospheric composition and OH reactivity in this region are under-represented, with this work providing important insights to the emissions and chemistry in this area. The paper is well written, within the scope of the journal, and will be of interest to the atmospheric science community. I recommend publication once the following minor points have been ad-

dressed:

Page 1, line 30: Please state the measurement uncertainty. Does this result imply that oxidation products of primary emissions have been measured in this campaign or that contributions to the OH reactivity from oxidation products is low in this region?

Page 4, line 15: Is there any impact of the length or the heating of the inlet on the trace gases sampled? While wall losses of gases may be reduced by heating, are there any other effects that need to be considered? Have any experiments been performed to test that the length and heating of the sampling line do not impact the concentrations of VOCs?

Page 4, line 28: Is the ratio of pyrrole to OH sufficiently high to ensure pseudo-first-order conditions?

Page 4, line 31: What was the variability in the measurements of C1?

Page 5, line 5: Please provide a summary of the VOC measurements and limits of detection.

Page 5, line 20: What is the typical percentage correction in the total reactivity? For the maximum correction of 7.4 s-1, what was the total reactivity?

Page 5, line 26: Please provide some further details explaining the basis for the corrections.

Page 6, lines 8-13: Please consider providing supplementary information which includes further details of which species were calibrated using gas standards and which masses have estimated concentrations.

Page 6, line 12: 'Formulas' to 'formulae'.

Page 6, section 2.5: Can an estimate of the total uncertainty in the calculated reactivity be provided?

Page 10, lines 12-19: The use of the ratio between C3 carbonyls and propane to determine the age of the air mass must assume there are no direct emissions of C3 carbonyls. Please provide some comments on the validity of this assumption.

Page 14, section 3.4: Does the extent of missing reactivity correlate with the estimated age of the air masses?

---

## Referee Comment (RC2) · Anonymous Referee #2 · 28 Jun 2019

General Comments

As is an important region for global change, the Arabian Peninsula is an area with intense photochemistry caused by strong solar radiation and strong emissions from the sea traffic, petrol chemical industry and the urban centers. Nevertheless, the Arabian Peninsula is in general lack of measurements of NOx, VOCs and the total OH reactivity. To fill this gap, a ship campaign with comprehensive suite of atmospheric measurements was conducted and the in-situ OH reactivity measurements was shown systemically to be a very useful parameters to characterize the impact of the anthropogenic emissions. Overall, I recommend publication at ACP after the authors have

addressed the following comments.

Specific Comments

1. Page 5 L15, what is the exact frequency to change the PTFE filter? It should be stated.

2. Page 5 L17-21, The interference by NO, NO2 and O3 toward the total OH reactivity (kOH) measurements determined in the lab on average seems to be quite different from those of the total OH reactivity comparison experiments presented by Fuchs et al., 2017. Details of the lab experiments shall be presented in this paper (maybe partly in the supporting materials) so that the measurement quality of the data is with higher transparency. The highest correction 7.4 s-1 at high NOx is actually comparable to the observed values. In this case, it might be useful to present a frequency distribution of the corrected kOH as the high NOx conditions seems to be frequently encountered during this ship campaign as shown by Figure 1.

3. Page 5 L23-25, are these test hydrocarbons representative for the air masses characterized in this study?

4. Page 10 L17, I don't quite understand how can be the ratio of the C3 carbonyls toward that of propane can be used as an indicator of oxidation state. The C3 carbonyls can be produced from larger NMHCs from propane and the degradation products can be C2 and C1 carbonyls. As the lifetime of C3 carbonyls and propane is different by one order of magnitude, the ratio is not good to characterize the oxidation state. It is probably still better to relay on the ratio of toluene to benzene or xylene to toluene for the analysis of the oxidation state.

5. Page 10 L29-30, what is the detection limit of the used total OH reactivity instrument?

6. Sect. 3.5.1 the ozone production regime is analyzed the ratio of the OH reactivities of VOCs to that of NOx. This is plausible but not accurate enough as the primary

production rate of ROx radicals will influence the ozone production regime as well. And since the NO2/NO ratio may be different for different VOCs/NOx ratio, the ozone production regime could still be different even under same VOCs/NOx ratio. So I suggest the influence of the ROx primary production rates (e.g. O3, HONO and carbonyls photolysis, etc) as well as the influence of the NO2/NO ratio shall be further analyzed.
* * *

---

## Author Comment (AC1) · 17 Jul 2019

Dear Referee,

Thank you very much for your constructive and helpful comments. We addressed them in the new version of the manuscript (attached with tracked changes) and below (changes in green):

**Page 1, line 30: Please state the measurement uncertainty. Does this result imply that oxidation products of primary emissions have been measured in this campaign or that contributions to the OH reactivity from oxidation products is**

[Figure]

**low in this region?**

It is a good idea to mention the uncertainty early in the manuscript, so we added it to the abstract. The contribution of oxidized species was measured as comprehensively as possible by PTR-ToF-MS and was relatively large, as we state earlier in the abstract: "Due to the rapid oxidation of direct volatile organic compound (VOC) emissions, oxygenated volatile organic compounds (OVOCs) were observed to be the main contributor to OH reactivity around the Arabian Peninsula ($9-35$ % by region)."

Changes made: We now report the "*measurement uncertainty (50%)*".

**Page 4, line 15: Is there any impact of the length or the heating of the inlet on the trace gases sampled? While wall losses of gases may be reduced by heating, are there any other effects that need to be considered? Have any experiments been performed to test that the length and heating of the sampling line do not impact the concentrations of VOCs?**

While no inlet loss tests were conducted with this specific inlet, we discuss in Sect 3.4 the probable loss of larger, less volatile VOCs in the inlet line due to its length. A further sentence on this topic has now been added to the methods section as well.

Changes made: "*We assume that the length of the inlet prevented some of the larger, less volatile VOCs from reaching the instruments (see Sect. 3.4).*"

**Page 4, line 28: Is the ratio of pyrrole to OH sufficiently high to ensure pseudo-first-order conditions?**

No, it is not, as it never is in the CRM approach to OH reactivity. This is why we performed a correction for deviation from pseudo-first-order conditions as described in section 2.4. In order to make this clearer we added the following sentence on p. 4, line 31: "*Because this ratio deviates from pseudo-first order conditions, as is typical for*

*CRM, a correction needed to be applied (see Sect. 2.4)."*

**Page 4, line 31: What was the variability in the measurements of C1?**

To clarify the variability observed in C1, we added the average and standard deviation of C1 measurements to the sentence: *"C1 was determined every 5−7 days (average ± standard deviation over the whole campaign: 60.71 ± 1.18 ppb of pyrrole)."*

**Page 5, line 5: Please provide a summary of the VOC measurements and limits of detection.**

We added this information to the manuscript in Sect. 2.3: *"Average detection limits for the different NMHCs ranged from 1−25 ppt. The NMHC and methane measurements are described in detail elsewhere (Bourtsoukidis et al., 2019). The PTR-ToF-MS was deployed at 60 °C drift temperature, 2.2 mbar drift pressure and 600 V drift voltage (E/N = 137 Td). 1,3,5-trichlorobenzene was fed continuously into the sample stream for mass scale calibration. The time resolution of the measurement was 1 min and background measurements were performed every 3 h for 10 min. The PTR-ToF-MS was calibrated with a multicomponent pressurized gas VOC standard (Apel-Riemer Environmental Inc., Colorado, USA). Mass resolution (full width at half maximum) at 96 amu ranged between ~3500 and ~4500. Average detection limits for the compounds measured by PTR-ToF-MS ranged from 1−107 ppt. A full list of the trace gases measured by GC-FID and PTR-ToF-MS can be found in Table S1."*

**Page 5, line 20: What is the typical percentage correction in the total reactivity? For the maximum correction of 7.4 s-1, what was the total reactivity?**

We made the following addition to the manuscript and added a Figure S1 (attached): *"The combined NO, NO2and ozone corrections amounted to a median of ~24 % of total*

*OH reactivity or an average of 0.36 ± 0.19 ppb of pyrrole over the whole campaign, which equals ≈ 1.9 s⁻¹, with a maximum absolute correction of 1.39 ppb (7.4 s⁻¹ / 27 % of total OH reactivity) under the influence of closely located ship emissions (high NOx). A frequency distribution of the corrected fraction of total OH reactivity is displayed in Fig. S1."*

**Page 5, line 26: Please provide some further details explaining the basis for the corrections.**

As requested we have now added further details explaining the necessity of the pseudo-1-st-order correction and hope this explains it sufficiently:

*"The calculation of total OH reactivity with the CRM equation assumes pseudo-1st-order conditions inside the reactor, which can, however, not be met while preserving a reasonable sensitivity (C3−C2 difference). The correction for deviation from pseudo-1st-order conditions inside the reactor was derived from empirical test gas measurements adapted from the approach in Michoud et al. (2015). The deviation from pseudo-1st-order depends on the pyrrole/OH ratio and on the reaction rate constant of the measured trace gas towards OH. Especially, trace gases with fast reaction rates (comparable to that of pyrrole) require the largest correction because their concentration inside the reactor decreases at a similar rate as that of pyrrole, leading to an underestimation of their reactivity. Such gases, namely alkenes, were present during AQABA (e.g. a C5H8-alkene which was not isoprene). This is why we did not use an average correction factor for all gas mixes as in Michoud et al. (2015), but we applied a correction where the composition of the air mass is taken into account.Test gases used were toluene (for aromatics correction), isoprene and propene (for alkenes correction), propane and cyclohexane (correction for other species). These test gases are representative of the relevant trace gases and ranges of reaction rate coefficients observed during AQABA. In test experiments, these trace gases were added to the CRM system in known amounts, i.e. known total OH reactivities, at pyrrole/OH ratios*

*representative for the campaign. The correction factors resulting from comparing the observed reactivity with the expected reactivity were weighted according to the measured composition of the ambient air at the moment of observation (. . .)"*

Full details of the method are available in the cited papers.

**Page 6, lines 8-13: Please consider providing supplementary information which includes further details of which species were calibrated using gas standards and which masses have estimated concentrations.**

We agree that this is important information and therefore included it into our Supplementary Information in Table S1 even in the initial submission: As indicated at the top of the table, grey shaded compounds were calibrated with gas standards and non-shaded compounds were calibrated with the theoretical approach. To make this clearer, we now added "specified in Table S1" to the text: *"Out of a total of 120 chemical species that were considered for the calculation of speciated reactivity, 42 chemical species (specified in Table S1), including many of the known important contributors to OH reactivity, were calibrated with gas standards and therefore have low uncertainties in their concentrations as well as in their reaction rate coefficients ($5-15$ %). A further 78 exact masses (specified in Table S1) monitored by $PTR-ToF-MS$ were attributed to molecular formulae "*

**Page 6, line 12: 'Formulas' to 'formulae'.**

We changed this in the text.

**Page 6, section 2.5: Can an estimate of the total uncertainty in the calculated reactivity be provided?**

This information is already given in this section: "*The uncertainty of the resulting speciated (i.e. calculated)OH reactivity depends on the fraction of gas-standard calibrated*

*compounds in the ambient air at any given point of time and varies between 10 % and 92 %, with an average and median of 45 % over the whole campaign.*" To further clarify that "speciated" reactivity is the same as calculated reactivity, we added *"(i.e. calculated)"* behind the word "speciated".

**Page 10, lines 12-19: The use of the ratio between C3 carbonyls and propane to determine the age of the air mass must assume there are no direct emissions of C3 carbonyls. Please provide some comments on the validity of this assumption.**

We agree that we cannot exclude direct emissions of carbonyls. However, when they are emitted together with propane, due to their longer lifetime they will remain present longer, contributing to the ubiquitous carbonyl "background" concentration and therefore later causing a larger C3 carbonyls/propane ratio. Due to the lifetime difference, we still expect a larger ratio in aged air masses than in freshly emitted ones.

Following the reviewer's comment we decided that a single ratio is not enough to indicate the oxidation state of an air mass. Therefore have added two more indicators (toluene/benzene and C5 carbonyls/pentanes). These both show the same general trends from more oxidized to less oxidized, with short sections where some direct emissions are apparent; see Table 1 and this addition to the text: *"The C3 carbonyls (C3H6O) vs. propane (C3H8), C5 oxidation products (C5H10O) vs. pentanes (C5H12) and toluene vs. benzene ratios (Table 1) are used here as an indicator of the oxidation state (and, thereby, the extent of photochemical ageing) of the air. We note that this approach is based on the assumption that production of the carbonyls by oxidation dominates over direct emission, although some direct combustion emissions cannot be excluded. The toluene vs. benzene relationship is based on the assumption of simultaneous emission of both compounds. Notably, a distinct difference between northern (less oxidized, [C3H6O]/[C3H8] ≈ 1.9 and [C5H10O]/[C5H12] ≈ 1.3) and southern Red Sea (oxidized, [C3H6O]/[C3H8] ≈ 9.4 and [C5H10O]/[C5H12] ≈ 1.9) was observed."*

**Page 14, section 3.4: Does the extent of missing reactivity correlate with the estimated age of the air masses?**

No, there is no trend visible (neither in fraction nor in absolute missing reactivity). We added this information to the text: "*
[revised manuscript text omitted]
$_3$H$_6$O) vs. propane (C$_3$H$_8$), $C_5$ oxidation products (C$_5$H$_{10}$O) vs. pentanes (C$_5$H$_{12}$) and toluene vs. benzene ratios  (Table 1) were used here as  indicators of the oxidation state (and, thereby, the extent of photochemical ageing) of the air. We note that this approach is based on the assumption that production of the carbonyls by oxidation dominates over direct emission, although some direct combustion emissions cannot be excluded. The toluene vs. benzene relationship is based on the assumption of simultaneous emission of both compounds. Notably, a distinct difference between northern (less oxidized,  [C$_3$H$_6$O]/[C$_3$H$_8$] ≈ 1.9 and [C$_5$H$_{10}$O]/[C$_5$H$_{12}$] ≈ 1.3) and southern Red Sea (oxidized [C$_3$H$_6$O]/[C$_3$H$_8$] ≈ 9.4 and [C$_5$H$_{10}$O]/[C$_5$H$_{12}$] ≈ 1.9) 
[revised manuscript text omitted]

---

## Author Comment (AC2) · 17 Jul 2019

Dear Referee,

We thank you for your constructive and helpful comments. They are addressed below and in the new version of the manuscript (attached together with the updated Supplement):

**Page 5 L15, what is the exact frequency to change the PTFE filter? It should be stated.**

The frequency has been added to the text: "*A PTFE (polytetrafluoroethylene) filter,*

[Figure]

*changed weekly (and additionally after intense particle contamination events such as dust storms), prevented contamination of the sub-sampling line by particles or sea spray.*"

**2. Page 5 L17-21, The interference by NO, NO2 and O3 toward the total OH reactivity (kOH) measurements determined in the lab on average seems to be quite different from those of the total OH reactivity comparison experiments presented by Fuchs et al., 2017. Details of the lab experiments shall be presented in this paper (maybe partly in the supporting materials) so that the measurement quality of the data is with higher transparency. The highest correction 7.4 s-1 at high NOx is actually comparable to the observed values. In this case, it might be useful to present a frequency distribution of the corrected kOH as the high Nox conditions seems to be frequently encountered during this ship campaign as shown by Figure 1.**

The corrections for NOx and ozone are, for the same pyrrole/OH ratios and NOx/ozone concentrations, comparable to those during the intercomparison described in Fuchs et al. (2017). However, the conditions during AQABA were usually different from that chamber based study, which naturally leads to differences in the resulting corrections. Ozone values were in the Arabian Gulf larger than ever tested during the intercomparison. A frequency distribution of the fraction of total OH reactivity due to this correction can now be found in Fig. S1 (attached). A description of the laboratory experiments has been added to the Supplementary Information as well. The following additions to the text were made:

*"Ozone, NO and NO2 interferences were quantified in the laboratory (details of the experiments can be found in Sect. S1). NO2 loss and conversion to NO in the mass flow controller during tests was monitored by an NO/NO2 analyzer (Two-channel-chemiluminescence detector). The combined NO, NO2 and ozone corrections amounted to a median of ˜24 % of total OH reactivity or an average of 0.36 ± 0.19 ppb*

*of pyrrole over the whole campaign, which equals $\approx$ 1.9 s$^{-1}$, with a maximum absolute correction of 1.39 ppb (7.4 s$^{-1}$ / 27 % of total OH reactivity) under the influence of closely located ship emissions (high NOx). A frequency distribution of the corrected fraction of total OH reactivity is displayed in Fig. S1."*

The following addition was made to the Supplement:

" S1. Laboratory characterizations of NO, NO2 and ozone interferences

S1.1 NO interference

To quantify the NO interference, NO from a standard gas cylinder (Air Liquide, Krefeld, Germany) was added to the CRM system running in C2 mode at pyrrole/OH $\approx$ 2.0. The mixing ratios of the added NO were ~0, 5, 10, 15, 20, 30, 90, 140, 170, 220, 350 ppb. The resulting decrease in pyrrole mixing ratio ($\Delta$C3 in ppb) vs. the NO mixing ratio resulted in a polynomial relationship of the form $\Delta$C3 = a[NO]$^2$ + b[NO]. During AQABA: a $\approx$ -3E-4; b $\approx$ 0.17.

S1.2 NO2 interference

To quantify the NO2 interference, NO2 from a standard gas cylinder (Westfalen AG, Münster, Germany) was added to the CRM system running in C2 mode at pyrrole/OH $\approx$ 2.0. All tubes were flushed with NO2 for several hours and the contact of the NO2 with metal surfaces (pressure reducer, mass flow controller) was reduced to a minimum to avoid losses. To quantify the amount of NO2 lost/converted to NO in the setup used, conversion to NO was monitored by an NO/NO2 analyzer (Two-channel-chemiluminescence detector). $8-12$ % of the NO2 was converted to NO depending on the flows and mixing ratios used. This loss was corrected for when calculating the NO2 mixing ratios in the reactor. The mixing ratios of the added NO2 were ~0, 20, 40, 60, 100 ppb. The resulting decrease in pyrrole mixing ratio ($\Delta$C3 in ppb) vs. the NO2 mixing ratio resulted in a polynomial relationship of the form $\Delta$C3 = a[NO2]$^2$ + b[NO2].

During AQABA: a $\approx$ -2E-4; b $\approx$ 0.08.

S1.3 Ozone interference

Using an ozone generator (Thermo 49C O3 Calibrator, Thermo Environmenal Instruments LLC, Franklin, USA), ozone was added to the CRM system running in C2 mode at pyrrole/OH $\approx$ 2.0. Ozone mixing ratios used were ~ 0, 20, 30, 45, 90, 130, 180, 260, 350 ppb. The resulting decrease in pyrrole mixing ratio ($\Delta$C3 in ppb) vs. the O3 mixing ratio resulted in a linear relationship of the form $\Delta$C3 = a[O3]. During AQABA: a $\approx$ 0.0058. "

**3. Page 5 L23-25, are these test hydrocarbons representative for the air masses characterized in this study?**

We agree that it is important to use test hydrocarbons that are relevant for the air masses seen in the study area. This is why we carefully evaluated the use of the available gas standards for calculating the correction factor. It is especially important to cover the relevant range of reaction rate constants of trace gases with OH, because this strongly impacts the correction for deviation from pseudo first order. Therefore we included isoprene despite its low abundance in the area, as some alkenes with fast reaction rate constants towards OH were observed. The explanation of this correction was extended in the manuscript:

*"The calculation of total OH reactivity with the CRM equation assumes pseudo-1st-order conditions inside the reactor, which can, however, not be met while preserving a reasonable sensitivity (C3− C2 difference). The correction for deviation from pseudo-1st-order conditions inside the reactor was derived from empirical test gas measurements adapted from the approach in Michoud et al. (2015). The deviation from pseudo-1st-order depends on the pyrrole/OH ratio and on the reaction rate con-*

*stant of the measured trace gas towards OH. Especially, trace gases with fast reaction rates (comparable to that of pyrrole) require the largest correction because their concentration inside the reactor decreases at a similar rate as that of pyrrole, leading to an underestimation of their reactivity. Such gases, namely alkenes, were present during AQABA (e.g. a C5H8-alkene which was not isoprene). This is why we did not use an average correction factor for all gas mixes as in Michoud et al. (2015), but we applied a correction where the composition of the air mass is taken into account.*Test gases used were toluene (for aromatics correction), isoprene and propene (for alkenes correction), propane and cyclohexane (correction for other species). *These test gases are representative of the relevant trace gases and ranges of reaction rate coefficients observed during AQABA. In test experiments, these trace gases were added to the CRM system in known amounts, i.e. known total OH reactivities, at pyrrole/OH ratios representative for the campaign. The correction factors resulting from comparing the observed reactivity with the expected reactivity* were weighted according to the measured composition of the ambient air at the moment of observation (. . .)"

Full details of the method are available in the cited papers.

**4. Page 10 L17, I don't quite understand how can be the ratio of the C3 carbonyls toward that of propane can be used as an indicator of oxidation state. The C3 carbonyls can be produced from larger NMHCs from propane and the degradation products can be C2 and C1 carbonyls. As the lifetime of C3 carbonyls and propane is different by one order of magnitude, the ratio is not good to characterize the oxidation state. It is probably still better to relay on the ratio of toluene to benzene or xylene to toluene for the analysis of the oxidation state**.

On reflection we agree that only one indicator for the oxidation state of the air mass is not enough. Therefore, we now added two more: toluene/benzene as suggested by the reviewer, and C5 carbonyls vs. pentanes. The latter were chosen because a far weaker contribution to C5 carbonyls via the fragmentation of higher hydrocarbons is expected

(see e.g. Master Chemical Mechanism), and the pentanes and their oxidation products have similar lifetimes. Even for C3, we consider the fragmentation a minor influence on the measured C3 carbonyls. We thank you for this suggestion because the two new ratios add information to the manuscript:

In Sect. 3.1.2: "*The C3 carbonyls vs. propane (C3H8), C5 oxidation products (C5H10O) vs. pentanes (C5H12) and toluene vs. benzene ratios (Table 1) were used here as indicators of the oxidation state (and, thereby, the extent of photochemical ageing) of the air. We note that this approach is based on the assumption that production of the carbonyls by oxidation dominates over direct emission, although some direct combustion emissions cannot be excluded. The toluene vs. benzene relationship is based on the assumption of simultaneous emission of both compounds. Notably, a distinct difference between northern (less oxidized, [C3H6O]/[C3H8] $\approx$ 1.9 and [C5H10O]/[C5H12] $\approx$ 1.3) and southern Red Sea (oxidized, [C3H6O]/[C3H8] $\approx$ 9.4 and [C5H10O]/[C5H12] $\approx$ 1.9) was observed.* "*

In Sect. 3.2.1:

*"Here, emission sources were closest to the point of observation, which was reflected in highest total OH reactivity (median: 18.8 s$^{-1}$, average $\pm$ standard deviation: 18.7 $\pm$ 7.9 s$^{-1}$) and low C3carbonyls/propane and high toluene/benzene ratio medians of 0.3 and 0.8, respectively. In contrast, OH reactivity in the air originating from Iran was lower (median: 9.8 s$^{-1}$, average $\pm$ standard deviation: 10.8 $\pm$ 3.4 s$^{-1}$), with a slightly higher C3carbonyls/propane and a lower toluene/benzene ratio (both 0.4).* "*

In Sect. 3.2.1:

*"The upper part of the graph shows indicators of the photochemical age of the air mass. The [C3H6O]/[C3H8] ratio decreases and the toluene/benzene ratio increases when getting closer to Kuwait, meaning that the research vessel was approaching the emission sources (refineries, oil platforms). Both these indicators show opposite patterns, as expected. The [C5H10O]/[C5H12] ratio follows the [C3H6O]/[C3H8] ratio*

*except for an increase towards Kuwait, which shows that the fresh emissions observed here were richer in propane than in pentanes, and indicates a mix of fresh emissions with aged air masses."*

In Sect. 3.2.2: *"The toluene/benzene ratio was between 1 and 3, indicating fuel combustion emissions as a source in Suez City and/or from ships, rather than biomass burning which tends to be richer in benzene."*

**5. Page 10 L29-30, what is the detection limit of the used total OH reactivity instrument?**

We added this information to the text: " *Total OH reactivity measured over the Arabian Sea during AQABA was with a median of 4.9 s$^{-1}$ (average $\pm$ standard deviation: 5.6 $\pm$ 2.0 s$^{-1}$) below the detection limit of 5.4 s$^{-1}$.*"

**6. Sect. 3.5.1 the ozone production regime is analyzed using the ratio of the OH reactivities of VOCs to that of NOx. This is plausible but not accurate enough as the primary production rate of ROx radicals will influence the ozone production regime as well.**

**And since the NO2/NO ratio may be different for different VOCs/NOx ratio, the ozone production regime could still be different even under same VOCs/NOx ratio. So I suggest the influence of the ROx primary production rates (e.g. O3, HONO and carbonyls photolysis, etc) as well as the influence of the NO2/NO ratio shall be further analyzed.**

The analysis of ozone production regimes as we present it here was developed by Kirchner et al. (2001). The authors of this method took into account the NO2/NO, HONO and ROx influences and extensively explain why the lifetime ratios of VOC/NOx (the inverse being daytime OH reactivities of NOx/VOC) are valid for distinguishing

between the different regimes. They also characterized the method in a modelling study which showed that this indicator of ozone production regimes performs better than other commonly used ones (e.g. NOy, O3/NOz). To make this clearer, we added the following to the text: *"The method and underlying chemistry has been extensively evaluated for different conditions and compared to other methods in Kirchner et al. (2001). The amount of peroxy radicals produced by VOC oxidation is linked to the OH reactivity of VOCs (…). Only data at daytime (06:00–18:00 local time) was considered, because the method inherently is limited to daytime chemistry."*

Thanks to your comment regarding the NO2/NO ratio, we realized that we can only use daytime values to assess the ozone production in a meaningful way. Therefore we have updated Fig. 9 (attached) accordingly.

---

## Author Response (AR2)

Dear Frank,

Thank you for this suggestion.

(Co-editor's comment: "In section 3.1.2 could you add a statement (p.10 line 20 or so) that in addition to the assumption of no direct emissions, the assumption is that the only precursor is the corresponding C3 and C5 compounds, i.e., no fragmentation of larger VOCs.")

We added the suggested statement to the text on page 10:

[revised manuscript text omitted]